# Free from Bellman Completeness: Trajectory Stitching via Model-based Return-conditioned Supervised Learning

**Zhaoyi Zhou[1], Chuning Zhu[2], Runlong Zhou[2], Qiwen Cui[2], Abhishek Gupta[2], Simon S. Du[2]**
[1] Institute for Interdisciplinary Information Sciences, Tsinghua University
[2] Paul G. Allen School of Computer Science and Engineering, University of Washington
`zhouzhao20@mails.tsinghua.edu.cn`
`{zchuning,vectorzh,qwcui,abhgupta,ssdu}@cs.washington.edu`

## Abstract

Off-policy dynamic programming (DP) techniques such as $Q$-learning have proven to be important in sequential decision-making problems. In the presence of function approximation, however, these techniques often diverge due to the absence of Bellman completeness in the function classes considered, a crucial condition for the success of DP-based methods. In this paper, we show how off-policy learning techniques based on return-conditioned supervised learning (RCSL) are able to circumvent these challenges of Bellman completeness, converging under significantly more relaxed assumptions inherited from supervised learning. We prove there exists a natural environment in which if one uses two-layer multilayer perceptron as the function approximator, the layer width needs to grow *linearly* with the state space size to satisfy Bellman completeness while a constant layer width is enough for RCSL. These findings take a step towards explaining the superior empirical performance of RCSL methods compared to DP-based methods in environments with near-optimal datasets. Furthermore, in order to learn from sub-optimal datasets, we propose a simple framework called MBRCSL, granting RCSL methods the ability of dynamic programming to stitch together segments from distinct trajectories. MBRCSL leverages learned dynamics models and forward sampling to accomplish trajectory stitching while avoiding the need for Bellman completeness that plagues all dynamic programming algorithms. We propose both theoretical analysis and experimental evaluation to back these claims, outperforming state-of-the-art model-free and model-based offline RL algorithms across several simulated robotics problems.[1]

## 1 Introduction

The literature in reinforcement learning (RL) has yielded a promising set of techniques for sequential decision-making, with several results showing the ability to synthesize complex behaviors in a variety of domains (Mnih et al., 2013; Lillicrap et al., 2015; Haarnoja et al., 2018) even in the absence of known models of dynamics and reward in the environment. Of particular interest in this work is a class of reinforcement learning algorithms known as off-policy RL algorithms (Fujimoto et al., 2018). Off-policy RL algorithms are those that are able to learn optimal behavior from an "off-policy" dataset - i.e. a potentially suboptimal dataset from some other policy. Moreover, off-policy RL methods are typically able to "stitch" together sub-trajectories across many different collecting trajectories, thereby showing the ability to synthesize behavior that is better than any trajectories present in the dataset (Degris et al., 2012; Kumar et al., 2019a).

The predominant class of off-policy RL algorithms builds on the idea of dynamic programming (DP) (Bellman, 1954; 1957) to learn state-action $Q$-functions. These techniques iteratively update the *Q-function* by applying the *Bellman operator* on the current $Q$-function. The step-wise update of

---

[1]Our code is available at `https://github.com/zhaoyizhou1123/mbrcsl`. Part of the work was done while Zhaoyi Zhou was visiting the University of Washington.

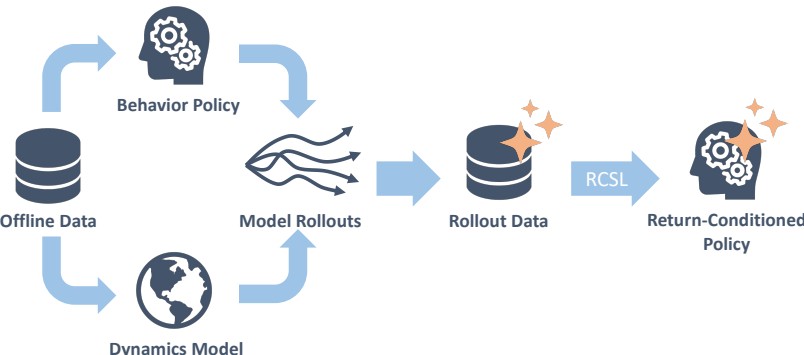

Figure 1: **Model-based return-conditioned supervised learning** (MBRCSL). Model-based rollouts augments datset with potentially optimal trajectories, enabling trajectory stitching for RCSL while avoiding Bellman completeness requirements.

DP enables the advantages listed above - usage of prior suboptimal data and synthesis of optimal policies by stitching together multiple suboptimal trajectories. On the other hand, these off-policy RL algorithms based on dynamic programming can only guarantee convergence under the condition of *Bellman completeness*, a strong and nuanced claim on function approximation that ensures the projection error during iterative DP can be reduced to be arbitrarily low. Unlike supervised learning, designing a function class that satisfies Bellman completeness is particularly challenging, as simply enlarging the function class does not guarantee Bellman completeness.

This begs the question - *can we design off-policy reinforcement learning algorithms that are free from the requirements of Bellman completeness, while retaining the advantages of DP-based off-policy RL?* To this end, we study a class of off-policy RL algorithms referred to as return-conditioned supervised learning (RCSL) (Schmidhuber, 2019; Srivastava et al., 2019). The key idea of RCSL is to learn a return-conditioned distribution of actions in each state by directly applying supervised learning on the trajectories in the dataset, achieving satisfactory performance by simply conditioning the policy on high desired returns. These techniques have seen recent prominence due to their surprisingly good empirical performance, simplicity and stability (Kumar et al., 2019b). While RCSL has empirically shown good results, the reason behind these successes in many empirical domains is still not well understood. In this work, we show how RCSL can circumvent the requirement for Bellman completeness. We theoretically analyze how RCSL benefits from not requiring Bellman completeness and thus can provably outperform DP-based algorithms with sufficient data coverage.

While RCSL is able to circumvent the requirement for Bellman completeness, it comes with the requirement for data coverage. As prior work has shown, RCSL has a fundamental inability to stitch trajectories (Brandfonbrener et al., 2022), a natural advantage of using DP-based off-policy RL. As long as the dataset does not cover optimal trajectories, RCSL methods are not guaranteed to output the optimal policy. To remedy this shortcoming, while preserving the benefit of avoiding Bellman completeness, we propose a novel algorithm - model-based RCSL (**MBRCSL**, cf. Figure 1). The key insight in **MBRCSL** is that RCSL style methods simply need data coverage, and can take care of optimizing for optimal behavior when the data provided has good (albeit skewed) coverage. Under the Markovian assumption that is made in RL, we show data coverage can be achieved by simply sampling from the behavior policy under a learned model of dynamics. We show how a conjunction of two supervised learning problems - learning a model of dynamics and of the behavior policy can provide us the data coverage needed for the optimality of RCSL. This leads to a method that both avoids Bellman completeness and is able to perform stitching and obtain the benefits of DP.

Concretely, the contributions of this work are:

1. We theoretically show how RCSL can outperform DP when Bellman completeness is not satisfied.
2. We theoretically show the shortcomings of RCSL when the off-policy dataset does not cover optimal trajectories.
3. We propose a novel technique - MBRCSL that can achieve trajectory stitching without suffering from the challenge of Bellman completeness.
4. We empirically show how MBRCSL can outperform both RCSL and DP-based off-policy RL algorithms in several domains in simulation.

## 1.1 RELATED WORK

**Function Approximation and Bellman Completeness.** Function approximation is required when the state-action space is large. A line of works has established that even if the optimal $Q$-function is realizable, the agent still needs exponential number of samples (Du et al., 2019; Wang et al., 2020; 2021; Weisz et al., 2021). Bellman completeness is a stronger condition than realizability but it permits sample-efficient algorithms and has been widely adopted in theoretical studies (Munos & Szepesvári, 2008; Chen & Jiang, 2019; Jin et al., 2020; Zanette et al., 2020; Xie et al., 2021).

**Return-conditioned supervised learning.** Return-conditioned supervised learning (RCSL) directly trains a return-conditioned policy via maximum likelihood (Schmidhuber, 2019; Srivastava et al., 2019; Kumar et al., 2019b; Andrychowicz et al., 2017; Furuta et al., 2021; Ghosh et al., 2019) The policy architecture ranges from MLP (Emmons et al., 2022; Brandfonbrener et al., 2022) to transformer (Chen et al., 2021; Bhargava et al., 2023) or diffusion models (Ajay et al., 2022).

The finding that RCSL cannot do stitching is first proposed by (Kumar et al., 2022) through empirical studies. Here, the trajectory stitching refers to combination of transitions from different trajectories to form potentially better trajectories, which is a key challenge to offline RL algorithms (Char et al., 2022; Hepburn & Montana, 2022; Wu et al., 2023). Brandfonbrener et al. (2022) gives an intuitive counter-example, but without a rigorous theorem. In Section 3.2, we provide two different reasons and rigorously show the the inability of RCSL to do stiching. Brandfonbrener et al. (2022) also derives a sample complexity bound of RCSL. However, they do not concretely characterize when RCSL requires much weaker conditions than $Q$-learning to find the optimal policy. Wu et al. (2023); Liu & Abbeel (2023); Yamagata et al. (2023) try to improve RCSL performance on sub-optimal trajectories by replacing dataset returns with (higher) estimated returns, while our approach aims at extracting *optimal* policy from dataset through explicit trajectory stitching. Besides, Paster et al. (2022); Yang et al. (2022) address the problems of RCSL in stochastic environments.

**Model-based RL.** Model-based RL learns an approximate dynamics model of the environment and extracts a policy using model rollouts (Chua et al., 2018; Janner et al., 2019; Hafner et al., 2021; Rybkin et al., 2021). A complementary line of work applies model-based RL to offline datasets by incorporating pessimism into model learning or policy extraction (Kidambi et al., 2020; Yu et al., 2020; 2021). We leverage the fact that models do not suffer from Bellman completeness and use it as a crucial tool to achieve trajectory stitching.

## 2 PRELIMINARIES

**Finite-horizon MDPs.** A finite-horizon Markov decision process (MDP) can be described as $\mathcal{M} = (H, \mathcal{S}, \mathcal{A}, \mu, \mathcal{T}, r)$, where $H$ is the planning horizon, $\mathcal{S}$ is the state space, and $\mathcal{A}$ is the action space. $\mu \in \Delta(\mathcal{S})$ is the initial state distribution, and $\mathcal{T} : \mathcal{S} \times \mathcal{A} \rightarrow \Delta(\mathcal{S})$ is the transition dynamics. Note that for any set $\mathcal{X}$, we use $\Delta(\mathcal{X})$ to denote the probability simplex over $\mathcal{X}$. In case of deterministic environments, we abuse the notation and write $\mathcal{T} : \mathcal{S} \times \mathcal{A} \rightarrow \mathcal{S}, r : \mathcal{S} \times \mathcal{A} \rightarrow [0, 1]$.

**Trajectories.** A trajectory is a tuple of states, actions and rewards from step 1 to $H$: $\tau = (s_1, a_1, r_1, \cdots, s_H, a_H, r_H)$. Define the return-to-go (RTG) of a trajectory $\tau$ at timestep $h$ as $g_h = \sum_{h'=h}^{H} r_{h'}$. The alternative representation of a trajectory is $\tau = (s_1, g_1, a_1; \cdots; s_H, g_H, a_H)$. We abuse the notation and call the sub-trajectory $\tau[i : j] = (s_i, g_i, a_i; \cdots; s_j, g_j, a_j)$ as a trajectory. Finally, we denote $\mathrm{T}$ as the set of all possible trajectories.

**Policies.** A policy $\pi : \mathrm{T} \rightarrow \Delta(\mathcal{A})$ is the distribution of actions conditioned on the trajectory as input. If the trajectory contains RTG $g_h$, then the policy is a *return-conditioned policy*. The Markovian property of MDPs ensures that the optimal policy falls in the class of *Markovian policies*, $\Pi^{\mathrm{M}} = \{\pi : \mathcal{S} \rightarrow \Delta(\mathcal{A})\}$. A policy class with our special interest is *Markovian return-conditioned policies*, $\Pi^{\mathrm{MRC}} = \{\pi : \mathcal{S} \times \mathbb{R} \rightarrow \Delta(\mathcal{A})\}$, i.e., the class of one-step return-conditioned policies.

**Value and $Q$-functions.** We define value and $Q$-functions for policies in $\Pi^{\mathrm{M}}$. Let $\mathbb{E}_\pi[\cdot]$ denote the expectation with respect to the random trajectory induced by $\pi \in \Pi^{\mathrm{MD}}$ in the MDP $\mathcal{M}$, that is, $\tau = (s_1, a_1, r_1, \ldots, s_H, a_H, r_H)$, where $s_1 \sim \mu(\cdot)$, $a_h \sim \pi(\cdot|s_h)$, $r_h = r^u(s_h, a_h)$, $s_{h+1} \sim \mathcal{T}(\cdot|s_h, a_h)$ for any $h \in [H]$. We define the value and $Q$-functions using the RTG $g_h$:

$$V_h^\pi(s) = \mathbb{E}_\pi [g_h | s_h = s], \ Q_h^\pi(s, a) = \mathbb{E}_\pi [g_h | s_h = s, a_h = a].$$

---

**Algorithm 1** Return-conditioned evaluation process

---

1: **Input:** Return-conditioned policy $\pi : \mathcal{S} \times \mathbb{R} \to \Delta(\mathcal{A})$, desired RTG $\tilde{g}_1$.
2: Observe $s_1$.
3: **for** $h = 1, 2, \cdots, H$ **do**
4:     Sample action $a_h \sim \pi(\cdot | s_h, \tilde{g}_h)$.
5:     Observe reward $r_h$ and next state $s_{h+1}$.
6:     Update RTG $\tilde{g}_{h+1} = \tilde{g}_h - r_h$.
7: **Return:** Actual RTG $g = \sum_{h=1}^{H} r_h$.

---

For convenience, we also define $V_{H+1}^{\pi}(s) = 0$, $Q_{H+1}^{\pi}(s, a) = 0$, for any policy $\pi$ and $s, a$.

**Expected RTG.** Return-conditioned policies needs an initial desired RTG $\tilde{g}_1$ to execute. The *return-conditioned evaluation process* for return-conditioned policies is formulated in Algorithm 1. We know that $\tilde{g}_1$ is not always equal to $g$, the real RTG. Hence to evaluate the performance of $\pi \in \Pi^{\text{MRC}}$, we denote by $J(\pi, \tilde{g}_1)$ the expected return $g$ under $\pi$ with initial desired RTG $\tilde{g}_1$.

**Offline RL.** In offline RL, we only have access to some fixed dataset $\mathcal{D} = \{\tau^1, \tau^2, \cdots, \tau^K\}$, where $\tau^i = (s_1^i, a_1^i, r_1^i, \cdots, s_H^i, a_H^i, r_H^i)$ $(i \in [K])$ is a trajectory sampled from the environment. We abuse the notation and define $(s, a) \in \mathcal{D}$ if and only if there exists some $i \in [K]$ and $h \in [H]$ such that $(s_h^i, a_h^i) = (s, a)$. We say dataset $\mathcal{D}$ *uniformly covers* $\mathcal{M}$ if $(s, a) \in \mathcal{D}$ for any $s \in \mathcal{S}$, $a \in \mathcal{A}$. For any deterministic optimal policy $\pi^* : \mathcal{S} \to \mathcal{A}$ of $\mathcal{M}$, we say dataset $\mathcal{D}$ *covers* $\pi^*$ if $(s, \pi^*(s)) \in \mathcal{D}$ for any $s \in \mathcal{S}$. The goal of offline RL is to derive a policy using $\mathcal{D}$ which maximizes the expected total reward. For Markovian policy $\pi \in \Pi^{\text{M}}$, the goal is $g^* = \max_{\pi} \sum_{s} \mu(s) V_1^{\pi}(s)$. For Markovian return-conditioned policy $\pi \in \Pi^{\text{MRC}}$, the goal is $\max_{\pi} J(\pi, g^*)$.

## 2.1   Realizability and Bellman Completeness of Q-Learning

Many exisiting RL algorithms, such as $Q$-learning and actor-critic, apply dynamic-programming (DP) technique (Bertsekas & Tsitsiklis, 1996) to estimate value functions or $Q$-functions. For instance, $Q$-learning updates estimated (optimal) $Q$-function $\widehat{Q}_h(s, a)$ by

$$\widehat{Q}_h \leftarrow \mathcal{B}\widehat{Q}_{h+1} = \{r(s, a) + \mathbb{E}_{s' \sim \mathcal{T}(\cdot | s, a)} \sup_{a'} \widehat{Q}_{h+1}(s', a')\}_{(s,a) \in \mathcal{S} \times \mathcal{A}}. \tag{1}$$

where $\mathcal{B}$ is called the *Bellman operator* (Sutton & Barto, 2018).

In practice, $\widehat{Q}$ is represented using some function approximation class such as multilayer perceptron (MLP). Two crucial conditions for DP-based methods with function approximation to succeed are realizability (Definition 1) and to be Bellman complete (Definition 2) (Munos, 2005; Zanette, 2023).

**Definition 1** (Realizability)**.** *The function class $\mathcal{F}$ contains the optimal $Q$-function: $Q^* \in \mathcal{F}$.*

**Definition 2** (Bellman complete)**.** *A function approximation class $\mathcal{F}$ is Bellman complete under Bellman operator $\mathcal{B}$ if $\max_{Q \in \mathcal{F}} \min_{Q' \in \mathcal{F}} \|Q' - \mathcal{B}Q\| = 0$.*

**Remark 1.** *It has been shown that realizability alone is not enough, as Weisz et al. (2021); Wang et al. (2020) show that it requires an exponential number of samples to find a near-optimal policy.*

## 2.2   Return-Conditioned Supervised Learning

RCSL methods learn a return-conditioned policy $\pi^{\text{cx}}(a_h | \tau[h - \text{cx} + 1 : h - 1], s_h, g_h)$ from $\mathcal{D}$ via supervised learning, where cx is the policy context. We focus on return-conditioned policies with context 1, i.e., policy of the form $\pi^1(a_h | s_h, g_h)$, because the optimal policy can be represented by a return-conditioned policy of context 1. Next, we introduce the *RTG dataset*, summarizes the information that RCSL algorithm relies on.

**Definition 3** (RTG dataset)**.** *Given a collection of trajectories $\mathcal{D} = \{\tau^1, \tau^2, \cdots, \tau^K\}$, an RTG dataset for $\mathcal{D}$ is defined by $\mathcal{D}^1 := \{(s_t^k, g_t^k, a_t^k) \mid 1 \le k \le K, 1 \le t \le H\}$.*

We are now ready to define the RCSL framework.

**Definition 4** (RCSL framework)**.** *An offline RL algorithm belongs to RCSL framework if it takes an RTG dataset as input and outputs a return-conditioned policy.*

# 3 UNDERSTANDING THE STRENGTHS AND WEAKNESSES OF RETURN-CONDITIONED SUPERVISED LEARNING METHODS

In this section, we first provide a theoretical discussion of why RCSL methods can outperform the DP methods in deterministic environments, when there is sufficient data coverage. We then give rigorous theoretical treatments to explain why RCSL style methods cannot do "trajectory-stitching".

## 3.1 INSIGHT 1: RCSL CAN OUTPERFORM DP IN DETERMINISTIC ENVIRONMENTS

In this section, we show that RCSL can achieve better performance than DP-based algorithms, as long as the dataset contains expert (optimal) trajectory. This stems from the fact that DP-algorithms needs to satisfy Bellman-completeness when function approximation is used, while RCSL algorithms only need to represent the optimal policy, and hence only require realizability.

To explain this point, we choose one representative implementation from both RCSL and DP-based methods respectively to make a comparison. For the former, we design "MLP-RCSL", a simple RCSL algorithm that learns a deterministic policy. Suppose that the action space $\mathcal{A}$ is a subset of $\mathbb{R}^d$. MLP-RCSL trains a policy network $\widehat{\pi}_\theta(s, g) \in \mathbb{R}^d$ by minimizing mean square error (MSE) of action (or more generally, maximize the log-likelihood):

$$L(\theta) = \mathbb{E}_{(s,g,a)\sim\mathcal{D}^1} \|\widehat{\pi}_\theta(s, g) - a\|^2. \tag{2}$$

We add a projection $f(a) = \arg\min_{a\in\mathcal{A}} \|a - \widehat{\pi}_\theta(s, g)\|$ at the output of policy network, so that we are ensured to get a valid action.

We choose $Q$-learning as a representative for DP-based methods, where the $Q$-function is learned by a $Q$-network, denoted by $\widehat{Q}_\phi(s, a)$. To make the comparison fair, both $\widehat{\pi}_\theta(s, g)$ and $\widehat{Q}_\phi(s, a)$ are implemented with a *two-layer* MLP network with ReLU activation.

**Remark 2.** *Often DP-based offline RL methods aim to penalize out-of-distribution actions (in the presence of incomplete coverage) via techniques such as conservativism (Kumar et al., 2020) or constrained optimization (Wu et al., 2019). However, we do not need to consider these techniques in our constructed environment, as conservatism is not needed when the dataset has full coverage.*

We first analyze when MLP-RCSL and $Q$-learning satisfy their own necessary requirements to succeed, in terms of the hidden layer size needed. To be specific, we set the requirements as:
- MLP-RCSL: The policy network must be able to represent the optimal policy.
- $Q$-learning: The $Q$-network must be able to represent the optimal $Q$-function and satisfy Bellman completeness.

**Theorem 1.** *There exists a series of MDPs and associated datasets $\{\mathcal{M}^u = (H^u, \mathcal{S}^u, \mathcal{A}, \mu, \mathcal{T}^u, r^u), \mathcal{D}^u\}_{u\geq 1}$, such that the following properties hold:*

*(i) $|\mathcal{S}^u| = \Theta(u)$;*
*(ii) $\mathcal{D}^u$ uniformly covers $\mathcal{M}^u$;*
*(iii) There exists a series of MLP-RCSL policy network (two-layer MLP) $\{\pi_{RC}^u\}_{u\geq 1}$ such that $\pi_{RC}^u$ can represent optimal policy for $\mathcal{M}^u$ with $O(1)$ neurons in the hidden layer;*
*(iv) For $u \geq 1$, any two-layer MLP Q-network representing the optimal Q-function and satisfying Bellman completeness simultaneously should have $\Omega(u)$ neurons in the hidden layer.*

We also provide simulations to verify the theorem (cf. A.1).

**Remark 3.** *From Theorem 1, we see that the model size of Q-learning must grow linearly with state space. On the other hand, RCSL can scale with large state space Besides, we also point out that the success of RCSL is based on the existence of expert trajectory.*

**Proof idea.** Detailed proof of Theorem 1 is deferred to Appendix A. We construct a class of MDPs called LinearQ (Figure 2), such that the optimal policy as well as optimal $Q$-function can be represented by the linear combination of a constant number of ReLU neurons, while the reward function must be represented with $\Omega(|\mathcal{S}^u|)$ ReLU neurons because there are that many. We denote the all-zero MLP as $\widehat{Q}_{\phi(0)}$ which satisfies $\widehat{Q}_{\phi(0)}(s, a) = 0$ for all $s \in \mathcal{S}^u$ and $a \in \mathcal{A}$. We see that Bellman operation (cf. Equation 1)) applies to the all-zero MLP give $\mathcal{B}\widehat{Q}_{\phi(0)}(s, a) = r^u(s, a), \forall s \in \mathcal{S}^u, a \in \mathcal{A}$. Thus $\Omega(|\mathcal{S}^u|)$ hidden neurons are needed to satisfy Bellman completeness.

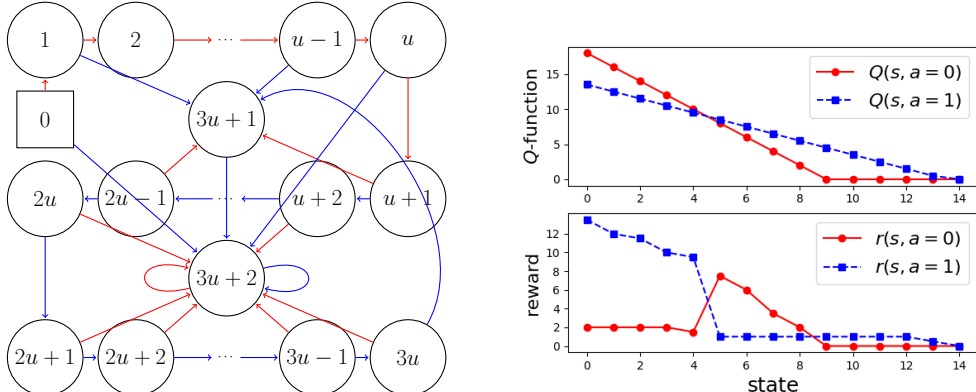

Figure 2: Left: Illustration for LinearQ transition dynamics. This figure is valid for $u$ being an even number. The transitions are plotted in red and blue arrows, where red arrows stand for $a = 0$ and blue ones for $a = 1$. The rewards are omitted. Upper right: Illustration for optimal $Q$-function of LinearQ with respect to states, where size parameter $u = 4$. Realizability can be satisfied with constant hidden neurons. Lower right: Illustration for reward function of LinearQ with respect to states, where size parameter $u = 4$. $\Omega(|\mathcal{S}^u|)$ hidden neurons are needed to represent reward.

### 3.2 INSIGHT 2: RCSL CANNOT PERFORM TRAJECTORY-STITCHING

It is generally believed that RCSL, cannot perform trajectory stitching (Brandfonbrener et al., 2022; Wu et al., 2023). We provide two theorems to rigorously show the sub-optimality of RCSL algorithms, even when the environment has *deterministic transition* and dataset satisfies *uniform coverage*. These two theorems present different rationales for the failure of trajectory stitching. Theorem 2 shows that Markovian RCSL (context length equal to 1) outputs a policy with a constant sub-optimal gap. Theorem 3 states that any decision transformer (using cross-entropy loss in RCSL) with any context length fails to stitch trajectories with non-zero probability.

**Theorem 2.** *For any Markovian RCSL algorithm, there always exists an MDP $\mathcal{M}$ and an (sub-optimal) offline dataset $\mathcal{D}$ such that*

    *(i) $\mathcal{M}$ has deterministic transitions;*
    *(ii) $\mathcal{D}$ uniformly covers $\mathcal{M}$;*
    *(iii) The output return-conditioned policy $\pi$ satisfies $|g^* - J(\pi, g^*)| \geq 1/2$ almost surely.*

The key idea is that the information of reward function cannot be recovered from the RTG dataset $\mathcal{D}^1$. We are able to construct two MDPs with different reward functions but the same RTG dataset. Therefore, at least RCSL will at least fail on one of them. The proof is deferred to Appendix B.1.

**Theorem 3.** *Under the assumption that a decision transformer outputs a mixture of policies for trajectories in the dataset for an unseen trajectory (details in Assumption 1), there exists an MDP $\mathcal{M}$ and an (sub-optimal) offline dataset $\mathcal{D}$ such that*

    *(i) $\mathcal{M}$ has deterministic transitions;*
    *(ii) $\mathcal{D}$ uniformly covers $\mathcal{M}$;*
    *(iii) Any decision transformer $\pi$ satisfies $J(\pi, g^*) < g^*$.*

The key idea is that cross-entropy loss of decision transformers forces non-zero probability for imitating trajectories in the dataset. We construct a dataset containing two sub-optimal trajectories with the same total reward, then their first step information $(s_1, g_1)$ are the same. We make their first step action $a_1$ and $a_1'$ different, so the RTG dataset contains two trajectories $(s_1, g_1, a_1)$ and $(s_1, g_1, a_1')$. Under the assumption on generalization ability to unseen trajectory (Assumption 1), the decision transformer will randomly pick actions with the first step $(s_1, g^* > g_1)$.

## 4 MODEL-BASED RETURN-CONDITIONED SUPERVISED LEARNING

In this section, we ask whether we can avoid Bellman-completeness requirements, while still being able to stitch together sub-optimal data. To this end, we propose model-based return-conditioned

---

**Algorithm 2** MBRCSL: Model-Based Return-Conditioned Supervised Learning

---

**Require:** Offline dataset $\mathcal{D}$, dynamics model $\widehat{T}_\theta$, behavior policy $\mu_\psi$ and output return-conditioned policy $\pi_\phi$, desired rollout dataset size $n$.

1: Collect initial states in $\mathcal{D}$ to form a dataset $\mathcal{D}_{\text{initial}}$. Compute the highest return $g_{\max}$ in $\mathcal{D}$.
2: Train dynamics model $\widehat{T}_\theta(s', r|s, a)$ and behavior policy $\mu_\psi(a|s)$ on $\mathcal{D}$ via MLE.
3: Initialize rollout dataset $\mathcal{D}_{\text{rollout}} \leftarrow \varnothing$, rollout dataset size counter $cnt \leftarrow 0$.
4: **for** $i = 1, 2, \cdots$ **do**
5:     Rollout a complete trajectory $\tau$ from $\mu_\psi$ and $\widehat{T}_\theta$, with initial state sampled from $\mathcal{D}_{\text{initial}}$.
6:     Compute the predicted total return $g$.
7:     **if** $g > g_{\max}$ **then**
8:         Add $\tau$ to $\mathcal{D}_{\text{rollout}}$.
9:         $cnt \leftarrow cnt + 1$.
10:         **if** $cnt == n$ **then** break
11: Train $\pi_\phi(a|s, g)$ on $\mathcal{D}_{\text{rollout}}$ via MLE.

---

supervised learning (MBRCSL) (cf. Algorithm 2). This technique brings the benefits of dynamic programming to RCSL without ever having to do dynamic programming backups, just forward simulation. As demonstrated in Section 3.1, when there is sufficient data coverage, RCSL can outperform DP, but typically this data coverage assumption does not hold.

The key idea behind MBRCSL is to augment the offline dataset with trajectories that themselves combine sub-components of many different trajectories (i.e "stitching"), on which we can then apply RCSL to acquire performance that is better than the original dataset. Since we are in the off-policy setting and want to avoid the requirement for additional environment access, we can instead learn an approximate dynamics model from the off-policy dataset, as well as an expressive multimodal behavior policy (Line 2), both using standard maximum likelihood estimation (MLE). Then, by simply rolling out (sampling i.i.d and executing sequentially) the behavior policy on the approximate dynamics model, we can generate "stitched" trajectories (Lines 4-10). In many cases, this generated data provides coverage over potentially optimal trajectory and return data, thereby enabling RCSL to accomplish better-than-dataset performance. We aggregate optimal trajectories into a new rollout dataset by picking out trajectories with returns larger than the maximum return in the dataset. Finally, applying RCSL on the augmented rollout dataset enables trajectory stitching without the need to satisfy Bellman completeness (Line 11). Another advantage of MBRCSL framework is its modular architecture, i.e., dynamics model, behavior policy and output policy architecture can be designed independently. This allows us to leverage the most performant model classes for each individual sub-component. The precise details of the architectures and optimization algorithms that empirically work best in each domain are left to the experiments (Section 5).

## 5 EXPERIMENTS

In our experimental evaluation, we are primarily focused on evaluating (a) how MBRCSL performs on offline RL tasks with sub-optimal datasets, in comparison with model-free and model-based offline RL baselines? Moreover, we hope to understand (b) How each component in MBRCSL affects overall performance? Question (b) is investigated in Appendix C with a complete ablation study.

### 5.1 EVALUATION ON POINT MAZE ENVIRONMENTS

We first evaluate the methods on a custom Point Maze environment built on top of D4RL Maze (Fu et al., 2020). As illustrated in Fig. 3, this task requires the agent to navigate a ball from the initial location (denoted $S$) to a designated goal (denoted $G$). To investigate if the methods can do stitching, we construct an offline dataset consisting of two kinds of suboptimal trajectories with equal number:
- A detour trajectory $S \rightarrow A \rightarrow B \rightarrow G$ that reaches the goal in a suboptimal manner.
- A trajectory for stitching: $S \rightarrow M$. This trajectory has very low return, but is essential for getting the optimal policy.

The optimal trajectory should be a stitching of the two trajectories in dataset ($S \rightarrow M \rightarrow G$). The trajectories are generated by a scripted policy similar to that in D4RL dataset (Fu et al., 2020). The resulting dataset has averaged return 40.7 and highest return 71.8.

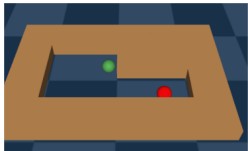 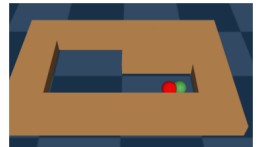 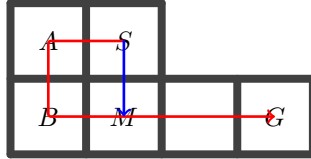

Figure 3: Point Maze illustration. Left and middle: Simulation images at initial and final timestep. Green point represents the ball to control, red point represents the target position. Right: Maze and dataset description. The initial point is $S$ and the goal is $G$. The dataset consists of two trajectories: $S \to A \to B \to G$ and $S \to M$.

To answer question (a), we compare MBRCSL with several offline RL baselines. (1) **CQL** (Kumar et al., 2020) is a model-free offline RL method that learns a conservative Q function penalizing high values for out of distribution actions. (2) **MOPO** (Yu et al., 2020) is a model-based offline RL method that learns an ensemble dynamics model and performs actor-critic learning using short-horizon model-rollouts. (3) **COMBO** (Yu et al., 2021) combines CQL and MOPO by jointly learning a conservative Q function and a dynamics model. (4) **MOReL** (Kidambi et al., 2020) first learns a pessimistic MDP from dataset and then learns a near-optimal policy in the pessimistic MDP. (5) **Decision Transformer (DT)** (Chen et al., 2021) is a RCSL method that leverages a transformer model to generate return-conditioned trajectories. (6) **%BC** performs behavior cloning on trajectories from the dataset with top 10% return.

We implement MBRCSL with the following architectures:

- Dynamics model learns a Gaussian distribution over the next state and reward: $\widehat{T}_\theta(s_{t+1}, r_t | s_t, a_t) = \mathcal{N}\left((s_{t+1}, r_t); \mu_\theta(s_t, a_t), \Sigma_\theta(s_t, a_t)\right)$. We learn an ensemble of dynamics models, in which each model is trained independently via maximum-likelihood. This is the same as many existing model-based approaches such as MOPO (Yu et al., 2020) and COMBO (Yu et al., 2021)
- We learn an approximate behavior policy with a diffusion policy architecture (Chi et al., 2023), which is able to capture multimodal action distributions. We fit the noise model by minimizing mean squared error (MSE).
- The return-conditioned policy is represented by a MLP network and trained by minimizing MSE.

As shown in Table 1, MBRCSL outperforms all baselines on the Point Maze environment, successfully stitching together suboptimal trajectories. COMBO, MOPO, MOReL and CQL struggles to recover the optimal trajectory, potentially due to challenges with Bellman completeness. They also have large variance in performance. In addition, we show in Appendix D that performance of DP-based offline RL methods cannot be improved by simply increasing model size. DT and %BC have low variance, but they cannot achieve higher returns than that of datset due to failure to perform stitching.

Table 1: Results on Point Maze. We record the mean and standard deviation (STD) on 4 seeds. The dataset has averaged return 40.7 and highest return 71.8.

| Task | MBRCSL (ours) | COMBO | MOPO | MOReL | CQL | DT | %BC |
|---|---|---|---|---|---|---|---|
| Pointmaze | **91.5±7.1** | 56.6±50.1 | 77.9±41.9 | 26.4±28.1 | 34.8± 24.9 | 57.2±4.1 | 54.0±9.2 |

## 5.2 Evaluation on Simulated Robotics Environments

We also evaluate the methods on three simulated robotic tasks from Singh et al. (2020). In each task, a robot arm is required to complete some goal by accomplishing two phases of actions, specified as follows:

- (**PickPlace**) (1) Pick an object from table; (2) Place the object into a tray.
- (**ClosedDrawer**) (1) Open a drawer; (2) Grasp the object in the drawer.
- (**BlockedDrawer**) (1) Close a drawer on top of the target drawer, and then open the target drawer; (2) Grasp the object in the target drawer.

We take images from Singh et al. (2020) for environment illustration (cf. Figure 4). For each task, the associated dataset consists of two kinds of trajectories: (i) Prior trajectories that only perform phase 1; (ii) Task trajectories which only perform phase 2, starting from the condition that phase 1 is completed. For PickPlace, ClosedDrawer and BlockedDrawer, the prior dataset completes phase 1 with probability about 40%, 70% and 90% respectively. The task dataset has success rate of about

90% for PickPlace and 60% for both ClosedDrawer and BlockedDrawer. The agent must stitch the prior and task trajectory together to complete the task. All three tasks have sparse reward, with return 1 for completing phase 2, and 0 otherwise. Comparing to Point Maze, the simulated robotic tasks require more precise dynamics estimates.

We choose **CQL** (Kumar et al., 2020), **COMBO** (Yu et al., 2021) and **Decision Transformer (DT)** (Chen et al., 2021) as baselines of model-free, model-based and RCSL methods, respectively. To demonstrate performance improvement of the output policy over the behavior policy in MBRCSL, we also tested behavior cloning implemented with diffusion policy (denoted by **BC**), which has the same architecture as the behavior policy in MBRCSL.

To capture the more complicated dynamics, we implemented the dynamics and output return-conditioned policy with the following architectures:

- Dynamics model is represented with a transformer model (Vaswani et al., 2017) $\widehat{T}_\theta(s', r | s, a)$, which is trained to minimize cross-entropy loss.
- The output return-conditioned policy is represented by an autoregressive model (Korenkevych et al., 2019) trained by MLE.

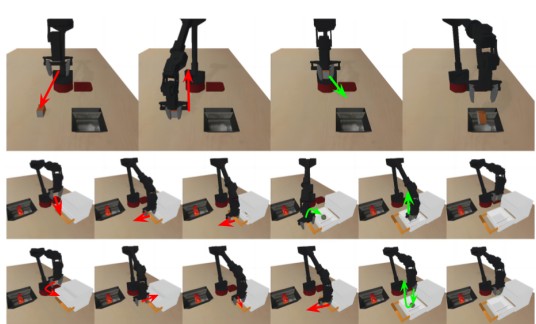

Figure 4: Simulated robotic tasks illustrations from CoG (Singh et al., 2020). The rows represent Pick-Place, ClosedDrawer and BlockedDrawer, respectively, from top to bottom. The dataset for each task consists of two kinds of trajectories: Some perform actions in red arrows and others perform actions in green arrows. A task is completed if and only if actions in both red and green arrows are performed successfully.

As shown in Table 2, MBRCSL outperforms all baseline methods in all three tasks. CQL achieves nonzero while low success rates and suffers from high variance in two of the tasks. This is possibly because Bellman completeness and sparse reward impede correct $Q$-function estimation. COMBO fails to complete the task at all, potentially due to inaccurate model rollouts incurring a higher $Q$ estimation error. DT fails to extract meaningful behavior because of a lack of expert trajectory. We also found that compared to the behavior policy (BC) which tries to capture all possible behaviors, the output policy (MBRCSL) is able to extract successful trajectories from rollouts and results in a higher success rate.

Table 2: Results on simulated robotic tasks. We record the mean and STD of the success rate on 4 seeds.

| Task | MBRCSL (ours) | CQL | COMBO | DT | BC |
|---|---|---|---|---|---|
| PickPlace | **0.48±0.04** | 0.22±0.35 | 0±0 | 0±0 | 0.07± 0.03 |
| ClosedDrawer | **0.51±0.12** | 0.11±0.08 | 0±0 | 0±0 | 0.38±0.02 |
| BlockedDrawer | **0.68±0.09** | 0.34±0.23 | 0±0 | 0±0 | 0.61±0.02 |

## 6 CONCLUSION

In this work, we conduct a thorough analysis of off-policy reinforcement learning, theoretically showing how return-conditioned supervised learning algorithms can provide a benefit over typical dynamic programming-based methods for reinforcement learning under function approximation through the viewpoint of Bellman completeness. We show that with data coverage, RCSL style algorithms do not require Bellman completeness, simply the easier realizability condition. We then characterize the limitations of RCSL in its inability to accomplish better-than-dataset behavior through trajectory stitching. To remedy this, while still retaining the benefit of avoiding Bellman completeness, we propose a novel algorithm - MBRCSL, that is able to accomplish data coverage through i.i.d forward sampling using a learned dynamics model and show its benefits. Notable open problems here concern how to make MBRCSL work in stochastic environments and scale these algorithms up to more complex and large-scale environments.

ACKNOWLEDGEMENTS

SSD acknowledges the support of NSF IIS 2110170, NSF DMS 2134106, NSF CCF 2212261, NSF IIS 2143493, NSF CCF 2019844, NSF IIS 2229881. CZ is supported by the UW-Amazon Fellowship.

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

# A    Supplementary Results for Section 3.1

## A.1    Counter-Example

**LinearQ Definition.** LinearQ is a class of MDPs, i.e., LinearQ $= \{\mathcal{M}^u \mid u \in \mathbb{N}^*\}$. Each MDP instance $\mathcal{M}^u$ of LinearQ class is associated with a "size parameter" $u$ and has state space $\mathcal{S}^u = \{0, 1, \cdots, 3u + 2\}$, action space $\mathcal{A} = \{0, 1\}$, and horizon $H^u = |\mathcal{S}^u| = 3u + 3$. The initial state is fixed to 0.

$\mathcal{M}^u$ has a deterministic transition function $\mathcal{T}^u : \mathcal{S}^u \times \mathcal{A} \to \mathcal{S}^u$, defined by

$$\mathcal{T}^u(s, a = 0) = \begin{cases} s + 1 & 0 \leq s \leq u \\ 3u + 2 & u + 1 \leq s \leq 2u \text{ and } s \text{ is even} \\ 3u + 1 & u + 1 \leq s \leq 2u \text{ and } s \text{ is odd} \\ 3u + 2 & 2u + 1 \leq s \leq 3u + 2, \end{cases}$$

$$\mathcal{T}^u(s, a = 1) = \begin{cases} 3u + 2 & 0 \leq s \leq u \text{ and } s \text{ is even} \\ 3u + 1 & 0 \leq s \leq u \text{ and } s \text{ is odd} \\ s + 1 & u + 1 \leq s \leq 3u + 1 \\ 3u + 2 & s = 3u + 2. \end{cases}$$

The reward function $r^u : \mathcal{S}^u \times \mathcal{A} \to [0, 1]$ is defined by

$$r^u(s, a = 0) = \begin{cases} 2k^u & 0 \leq s \leq u - 1 \\ 1.5k^u & s = u \\ (-2s + 4u + 2)k^u & u + 1 \leq s \leq 2u \text{ and } s \text{ is even} \\ (-2s + 4u + 1.5)k^u & u + 1 \leq s \leq 2u \text{ and } s \text{ is odd} \\ 0 & 2u + 1 \leq s \leq 3u + 2, \end{cases}$$

$$r^u(s, a = 1) = \begin{cases} (-s + 3u + 1.5)k^u & 0 \leq s \leq u \text{ and } s \text{ is even} \\ (-s + 3u + 1)k^u & 0 \leq s \leq u \text{ and } s \text{ is odd} \\ k^u & u + 1 \leq s \leq 3u + 1 \\ 0 & s = 3u + 2. \end{cases}$$

Here $k^u = \frac{1}{3u + 1}$ is used to scale the reward to range $[0, 1]$.

**Properties.** The name "LinearQ" comes from its $Q$-function, which can be represented by a ReLU function (Figure 2 right and Lemma 1).

**Lemma 1.** *For any MDP $\mathcal{M}^u \in$ LinearQ with size parameter $u$, the optimal $Q$-function is*

$$Q^{u,*}(s, a) = \begin{cases} 2k^u \cdot \text{ReLU}(-s + 2u + 1) & a = 0 \\ k^u \cdot \text{ReLU}(-s + 3u + 1.5) & a = 1. \end{cases}$$

*As a result, the optimal value function is*

$$V^{u,*}(s) = \begin{cases} Q^{u,*}(s, a = 0) = 2k^u(-s + 2u + 1) & 0 \leq s \leq u \\ Q^{u,*}(s, a = 1) = k^u(-s + 3u + 1.5) & u + 1 \leq s \leq 3u + 1 \\ 0 & s = 3u + 2, \end{cases}$$

*and the optimal policy $\pi^{u,*} : \mathcal{S}^u \to \mathcal{A}$ is*

$$\pi^{u,*}(s) = \begin{cases} 0 & 0 \leq s \leq u \\ 1 & u + 1 \leq s \leq 3u + 2 \end{cases} \tag{3}$$

*with optimal return $V^{u,*}(0) = 2k^u(2u + 1)$.*

*Proof.* We prove by induction on state $s$. When $s = 3u + 2$, the agent will always be in state $3u + 2$ with reward 0, so $Q^{u,*}(3u + 2, a) = 0$, for all $a \in \mathcal{A}$.

Now suppose $Q^{u,*}(n, 0) = 2k^u \cdot \text{ReLU}(-n + 2u + 1)$ and $Q^{u,*}(n, 1) = k^u \cdot \text{ReLU}(-n + 3u + 1.5)$ for some integer $n$ in $[1, 3u + 2]$. Then consider state $s = n - 1 \in [0, 3u + 1]$. For simplicity, we use $s'$ to denote the next state given state $s$ and action $a$. If $a = 0$, there are 5 possible cases:

- $0 \leq s \leq u - 1$. Then $s' = s + 1 \leq u$, and

$$Q^{u,*}(s,0) = r^u(s,0) + V^{u,*}(s') = r^u(s,0) + Q^{u,*}(s+1,0) = 2k^u(-s + 2u + 1).$$

- $s = u$. Then $s' = u + 1$, and

$$Q^{u,*}(s,0) = r^u(s,0) + V^{u,*}(s') = r^u(s,0) + Q^{u,*}(u+1,1) = 2k^u(-s + 2u + 1).$$

- $u + 1 \leq s \leq 2u$ and $s$ is even. Then $s' = 3u + 2$, and

$$Q^{u,*}(s,0) = r^u(s,0) + V^{u,*}(s') = r^u(s,0) + 0 = 2k^u(-s + 2u + 1).$$

- $u + 1 \leq s \leq 2u$ and $s$ is odd. Then $s' = 3u + 1$, and

$$Q^{u,*}(s,0) = r^u(s,0) + V^{u,*}(s') = r^u(s,0) + Q^{u,*}(3u+1,1) = 2k^u(-s + 2u + 1).$$

- $2u + 1 \leq s \leq 3u + 1$. Then $s' = 3u + 2$, and

$$Q^{u,*}(s,0) = r^u(s,0) + V^{u,*}(s') = r^u(s,0) + 0 = 0.$$

If $a = 1$, there are 3 possible cases:

- $0 \leq s \leq u$ and $s$ is even. Then $s' = 3u + 2$, and

$$Q^{u,*}(s,1) = r^u(s,1) + V^{u,*}(s') = r^u(s,1) + 0 = k^u(-s + 3u + 1.5).$$

- $0 \leq s \leq u$ and $s$ is odd. Then $s' = 3u + 1$, and

$$Q^{u,*}(s,1) = r^u(s,1) + V^{u,*}(s') = r^u(s,1) + Q^{u,*}(3u+1,1) = k^u(-s + 3u + 1.5).$$

- $u + 1 \leq s \leq 3u + 1$. Then $s' = s + 1 \geq u$, and

$$Q^{u,*}(s,1) = r^u(s,1) + V^{u,*}(s') = r^u(s,1) + Q^{u,*}(s+1,1) = k^u(-s + 3u + 1.5).$$

Therefore, $Q^{u,*}(s = n - 1, 0) = 2k^u \cdot \text{ReLU}(-(n-1) + 2u + 1)$ and $Q^{u,*}(s = n - 1, 1) = k^u \cdot \text{ReLU}(-(n-1) + 3u + 1.5)$. Thus we complete the proof by induction. $\square$

**Dataset.** The dataset $\mathcal{D}^u$ contains $4(3u + 3)n$ trajectories, where $n \in \mathbb{N}^*$ is a hyper-parameter for dataset size. $3(3u + 3)n$ trajectories are sampled by the optimal policy $\pi^{u,*}$. We further define $(3u + 3)$ "one-step-deviation" policy $\pi^{u,t}(s)$ ($t \in \mathcal{S}$), with each policy sampling $n$ trajectories:

$$\pi^{u,t}(s) = \begin{cases} \pi^{u,*}(s) & s \neq t \\ 1 - \pi^{u,*}(s) & s = t. \end{cases}$$

**Simulation Experiment** We test the performance of MLP-RCSL and $Q$-learning with various network size, and record the gap between achieved return and optimal return. During evaluation process of MLP-RCSL, we input the exact optimal return as desired RTG. We also present the performance of a naive policy as baseline, which always takes action 0 regardless of current state. Experiment details and hyperparameter choices are listed in Appendix E.3. As presented in Table 3, MLP-RCSL reaches optimal behavior with constant step size, while $Q$-learning's best performance only matches the naive policy, even if the network hidden layer size increases linearly with state space size.

## A.2 Proof for Theorem 1

Consider LinearQ (Appendix A.1) as our counterexample, then $(i)$ and $(ii)$ hold directly.

For the rest of the theorem, we need to first introduce some basic lemmas concerning the relationship of ReLU functions with indicator function in integer domain. The results below are easy to verify and thus we omit the proof.

Table 3: **Simulation results for LinearQ** with different size parameter $u$. "Naive" stands for the policy that always takes action 0. "RCSL-$w$" means MLP-RCSL with hidden layer size $w$, "QL-$w$" means $Q$-learning with hidden layer size $w$. We report the difference between achieved return on the last training epoch and the optimal return. RCSL achieves optimal return with 16 hidden layers, while $Q$-learning only achieves return same as a naive policy.

| $u$ | Naive | RCSL-16 | QL-16 | QL-0.5$u$ | QL-$u$ | QL-2$u$ | QL-4$u$ |
|---|---|---|---|---|---|---|---|
| 16 | 1 | 0 | 1 | 1 | 1 | 1 | 1 |
| 32 | 1 | 0 | 1 | 1 | 1 | 32.5 | 1 |
| 48 | 1 | 0 | 1 | 1 | 1 | 1 | 1 |
| 64 | 1 | 0 | 1 | 1 | 1 | 1 | 1 |
| 80 | 1 | 0 | 1 | 1 | 1 | 1 | 1 |
| 96 | 1 | 0 | 1 | 1 | 1 | 1 | 50.5 |
| 112 | 1 | 0 | 1 | 1 | 1 | 1 | 1 |
| 128 | 1 | 0 | 1 | 1 | 1 | 1 | 1 |
| 144 | 1 | 0 | 1 | 1 | 126 | 1 | 1 |
| 160 | 1 | 0 | 1 | 1 | 1 | 1 | 1 |

**Lemma 2.** *For any $x, u \in \mathbb{Z}$, we have*

$$\mathbf{1}_{x \leq u} = -\operatorname{ReLU}(-x + u) + \operatorname{ReLU}(-x + u + 1)$$
$$\mathbf{1}_{x \geq u} = -\operatorname{ReLU}(x - u) + \operatorname{ReLU}(x - u + 1)$$
$$\mathbf{1}_{x = u} = -\operatorname{ReLU}(-x + u) + \operatorname{ReLU}(-x + u + 1)$$
$$\quad -\operatorname{ReLU}(x - u) + \operatorname{ReLU}(x - u + 1) - 1.$$

*Proof of $(iii)$ in Theorem 1.* We first observe some critical facts of the dataset $\mathcal{D}^u$. For any state $s$, its sub-optimal action $1 - \pi^{u,*}(s)$ only appears in trajectory sampled by policy $\pi^{u,s}$, which achieves RTG $Q^{u,*}(s, 1 - \pi^{u,*}(s))$ at state $s$, since actions after state $s$ are all optimal. Thus a return-conditioned policy $\pi(s, g)$ of the form

$$\pi(s, g) = \begin{cases} \pi^{u,*}(s) & g \neq Q^{u,*}(s, 1 - \pi^{u,*}(s)) \\ 1 - \pi^{u,*}(s) & g = Q^{u,*}(s, 1 - \pi^{u,*}(s)) \end{cases}$$

achieves zero training error on $\mathcal{D}^u$.

Define $f(x) = \mathbf{1}_{x \geq 0.5}, \forall x \in \mathbb{R}$, which is the output projection function in MLP-RCSL. We show that

$$\pi(s, g) = f(\mathbf{1}_{g + s = 3u + 1.5} - \mathbf{1}_{g + s = 2u + 1} - 2\mathbf{1}_{g = 0} - \mathbf{1}_{s \leq u} + 1) \tag{4}$$

via case analysis on state $s$ (MLP only needs to represent the input to $f$):

- $0 \leq s \leq u$. Then the optimal action is $\pi^{u,*}(s) = 0$. We have $\mathbf{1}_{s \leq u} = 1$. In addition, we have $\mathbf{1}_{g = 0} = 0$, because the reward on state $s \in [0, u]$ is always positive and thus the RTG $g$ must be positive.

  Now if $g + s = 3u + 1.5$, then $\mathbf{1}_{g + s = 2u + 1} = 0$, and RHS of (4) is 1, which equals $\pi(s, g)$ as $g = Q^{u,*}(s, 1)$. If $g + s \neq 3u + 1.5$, then RHS of (4) is always 0, which is also the same as $\pi(s, g)$.

- $u + 1 \leq s \leq 2u + 1$. Then the optimal action is $\pi^{u,*}(s) = 1$. We have $\mathbf{1}_{s \leq u} = 0$. In addition, we have $\mathbf{1}_{g = 0} = 0$, because the reward on state $s \in [0, u]$ is always positive and thus the RTG $g$ must be positive.

  Now if $g + s = 2u + 1$, then $\mathbf{1}_{g + s = 3u + 1.5} = 0$, and RHS of (4) is 0, which equals $\pi(s, g)$ as $g = Q^{u,*}(s, 0)$. If $g + s \neq 3u + 1.5$, then RHS of (4) is always 1, which is also the same as $\pi(s, g)$.

- $2u + 2 \leq s \leq 3u + 2$. Then the optimal action is $\pi^{u,*}(s) = 1$. We have $\mathbf{1}_{s \leq u} = 0$. In addition, we have $\mathbf{1}_{g + s = 2u + 1} = 0$, because $g + s \geq g \leq 2u + 2$.

  Now if $g = 0$, then RHS of (4) is always 0, which equals $\pi(s, g)$ as $g = Q^{u,*}(s, 0)$. If $g \neq 0$, then RHS of (4) is always 1, which is also the same as $\pi(s, g)$.

Therefore, Equation (4) holds. By Lemma 2, each indicator function in (4) can be represented with 4 ReLU functions with input $s$ and $g$. As a result, we can achieve zero training error with 16 hidden neurons. □

We can show that the optimal $Q$-function can be represented by 2 hidden neurons. Notice that we have for all $s \in \mathcal{S}$ and $a \in \mathcal{A} = \{0, 1\}$ that

$$Q^{u,*}(s, a) = 2k^u \cdot \text{ReLU}(-s + 2u + 1) \cdot \mathbf{1}_{a=0} + k^u \cdot \text{ReLU}(-s + 3u + 1.5) \cdot \mathbf{1}_{a=1}$$
$$= 2k^u \cdot \text{ReLU}(-s - (2u + 1)a + 2u + 1) + k^u \cdot \text{ReLU}(-s + (3u + 1.5)a).$$

Thus $Q^{u,*}(s, a)$ can be represented by 2 hidden neurons, i.e. $\text{ReLU}(-s - (2u + 1)a + 2u + 1)$ and $\text{ReLU}(-s + (3u + 1.5)a)$.

*Proof of $(iv)$ in Theorem 1.* We just need to consider the case when $a = 0$. Notice that for $s \in [u + 1, 2u]$, we have

$$r^u(s, a = 0) = Q^{u,*}(s, a = 0) - V^{u,*}(\mathcal{T}^u(s, a = 0))$$
$$= \begin{cases} k^u(-2s + 4u + 2) & s \text{ is even} \\ k^u(-2s + 4u + 1.5) & s \text{ is odd}. \end{cases}$$

For any array $\{v(k)\}_{k=l_1}^{l_2}$ $(l_1 \le l_2 - 2)$, we recursively define the $t$-th order difference array of $\{v(k)\}_{k=l_1}^{l_2}$ $(t \le l_2 - l_1)$, denoted by $\{\Delta^{(t)}v(k)\}_{k=l_1}^{l_2-t}$, as follows:

$$\begin{cases} \Delta^{(t)}v(k) = \Delta^{(t-1)}v(k+1) - \Delta^{(t-1)}v(k), & \forall t \ge 1, l_1 \le k \le l_2 - t \\ \Delta^{(0)}v(k) = v(k), & \forall l_1 \le k \le l_2. \end{cases}$$

We fix $u$ and construct an array $\{R(k)\}_{k=u+1}^{2u}$ where $R(k) = r^u(s = k, a = 0)$ for $k \in \{u, u + 1, \cdots, 2u\}$. Then its second-order difference array is

$$\Delta^{(2)}R(k) = (-1)^k, \forall u + 1 \le k \le 2u - 2,$$

which has $u - 2$ nonzero terms.

On the other hand, the $Q$ network must be able to represent reward function in order to fulfill Bellman completeness, as is discussed in proof idea of Section 3.1. Suppose the $Q$-network has $m$ hidden neurons. Then it can represent functions of the form

$$f(s, a) = \sum_{i=1}^{m} w_i \text{ReLU}(\alpha_i s + \beta_i a + c_i) + c,$$

where $w_i, \alpha_i, \beta_i, c_i$ $(1 \le i \le m)$ as well as $c$ are all real numbers. Define array $\{\widehat{R}(k)\}_{k=u+1}^{2u}$ and a series of arrays $\{L_i(k)\}_{k=u+1}^{2u}$ $(1 \le i \le m)$, in which $\widehat{R}(k) = f(s = k, a = 0)$ and $L_i(k) = \text{ReLU}(\alpha_i k + c_i)$ for any $k \in \{u, u+1, \cdots, 2u), i \in [m]$. Then the second-order difference array of $\{\widehat{R}(k)\}_{k=u+1}^{2u}$ is

$$\Delta^{(2)}\widehat{R}(k) = \sum_{i=1}^{m} w_i \Delta^{(2)}L_i(k), \forall u + 1 \le k \le 2u - 2.$$

Notice that for any $i \in [m]$, there exists at most two $k$'s in $[u + 1, 2u + 2]$, such that $\Delta^{(2)}L_i(k) \ne 0$. This indicates that $\Delta^{(2)}R(k)$ has at most $2m$ non-zero terms.

To make $f(s, a) = r(s, a)$ for any $s \in \mathcal{S}^u$, $a \in \mathcal{A}$, we must have $\{R(k)\}_{k=u+1}^{2u} = \{\widehat{R}(k)\}_{k=u+1}^{2u}$, and thus $\{\Delta^{(2)}R(k)\}_{k=u+1}^{2u} = \{\Delta^{(2)}\widehat{R}(k)\}_{k=u+1}^{2u}$. Therefore, we have $2m \ge u - 2$, indicating that there are at least $\frac{u-2}{2} = \Omega(|\mathcal{S}^u|)$ hidden neurons. □

A.3 Experiment

We empirically validate the claim of Theorem 1 on an environment with continuous state and action space. To be specific, We compared RCSL and CQL on Point Maze task (Section 5.1) with expert dataset (referred to as "**Pointmaze expert**"). The expert dataset contains two kinds of trajectories: 1) $S \to A \to B$ and 2) $S \to M \to G$ (optimal trajectory). Please refer to Figure 3 for the maze illustration. The highest return in dataset is 137.9.

Both RCSL policy and CQL Q-network are implemented with three-layer MLP networks, with every hidden layer width equal to hyper-parameter "dim". We recorded the averaged returns on 4 seeds under different choices of 'dim'.

Table 4: Comparison of RCSL and CQL on Pointmaze expert under different model size. We recorded the averaged return on 4 seeds.

|  | dim=64 | dim=128 | dim=256 | dim=512 | dim=1024 |
|---|---|---|---|---|---|
| RCSL | 89.6 | 97.8 | 93.3 | 113 | 105 |
| CQL | 11.5 | 11.7 | 14.7 | 37.8 | 23 |

As shown in Table 4, CQL performance is much worse than RCSL on all model sizes. This is because Bellman completeness is not satisfied for CQL, even with a large model. In contrast, realizability of RCSL can be easily fulfilled with a relatively small model. The experiment supports the theoretical result of Theorem 1 that RCSL can outperform DP-based offline RL methods on deterministic environments, as RCSL avoids Bellman completeness requirements.

# B Proofs for Section 3.2

## B.1 Proof of Theorem 2

**Construction of MDPs.** Fix any integer constant $H_0 \geq 1$. Consider two MDPs $\mathcal{M}_1$, $\mathcal{M}_2$ with horizon $H = 2H_0$ specified as follows:

- State space: Both have only one state denoted by $s$.
- Action space: Both have two actions, $a_L$ and $a_R$.
- Transition: The transition is trivial as there is only one state.
- Reward: There are two possible rewards: good reward $r_g$ and bad reward $r_b$, where $r_g > r_b$. For $\mathcal{M}_1$, and all $h \in [H_0]$, we have

$$r_{2h,\mathcal{M}_1}(s, a_L) = r_g$$
$$r_{2h,\mathcal{M}_1}(s, a_R) = r_b$$
$$r_{2h-1,\mathcal{M}_1}(s, a_L) = r_b$$
$$r_{2h-1,\mathcal{M}_1}(s, a_R) = r_g.$$

  For $\mathcal{M}_2$, we have for all $h \in [2H_0]$

$$r_{h,\mathcal{M}_2}(s, a_L) = r_g$$
$$r_{h,\mathcal{M}_2}(s, a_R) = r_b.$$

**Construction of Dataset.** Fix any integer constant $K_0 \geq 1$. We construct datasets with $K = 2K_0$ trajectories for $\mathcal{M}_1$ and $\mathcal{M}_2$ respectively.

For $\mathcal{M}_1$, the trajectories are sampled by two policies: One policy always takes $a_L$ and the other always takes $a_R$. That is, there are two possible RTG-trajectories:

$$\tau_1^1 = (s, H_0\overline{r}, a_L, s, (H_0 - 1)\overline{r} + r_g, a_L, \cdots, s, r_g, a_L)$$
$$\tau_1^2 = (s, H_0\overline{r}, a_R, s, (H_0 - 1)\overline{r} + r_b, a_R, \cdots, s, r_b, a_R).$$

Let $\mathcal{D}_1$ be the dataset containing $K_0$ $\tau_1^1$'s and $K_0$ $\tau_1^2$'s, and $\mathcal{D}_1^1$ be the resulting trajectory-slice dataset with context length 1 for $\mathcal{D}_1$.

For $\mathcal{M}_2$, the trajectories are also sampled by two policies. Both policies alternate between actions $a_L$ and $a_R$, one starting with $a_L$ and the other starting with $a_R$. Then there are two possible RTG-trajectories:

$$\tau_2^1 = (s, H_0\bar{r}, a_L, s, (H_0 - 1)\bar{r} + r_b, a_R, \cdots, s, r_b, a_R)$$
$$\tau_2^2 = (s, H_0\bar{r}, a_R, s, (H_0 - 1)\bar{r} + r_g, a_L, \cdots, s, r_g, a_L).$$

Let $\mathcal{D}_2$ be the dataset containing $K_0$ $\tau_2^1$'s and $K_0$ $\tau_2^2$'s, and $\mathcal{D}_2^1$ be the resulting trajectory-slice dataset with context length 1 for $\mathcal{D}_2$.

It is obvious that the datasets are compliant with their corresponding MDP. We also point out that $\mathcal{D}_1^1 = \mathcal{D}_2^1$. In other words, the trajectory-slice datasets with context length 1 for $\mathcal{M}_1$ and $\mathcal{M}_2$ are exactly the same.

**Optimal returns and policies.** For both MDPs, the optimal expected RTG is $g^* = Hr_g$, and the optimal policy is to take action with reward $r_g$ at every step.

**Analysis of RCSL algorithms** Consider Any RCSL algorithm with context length 1. Suppose it outputs $\pi^1$, an RCSL policy with context length 1, given input $\mathcal{D}_1^1 = \mathcal{D}_2^1$. At step 1, the conditioned RTG is $g^*$ for both $\mathcal{M}_1$ and $\mathcal{M}_2$. Suppose $\pi^1$ takes $a_L$ with probability $p$ and $a_R$ with probability $1-p$ when conditioning on $g^*$, where $0 \le p \le 1$. Then the expected reward at step 1 is $pr_b + (1-p)r_g$ in $\mathcal{M}_1$ and $pr_g + (1-p)r_b$. Since

$$[pr_b + (1-p)r_g] + [pr_g + (1-p)r_b] = r_b + r_g,$$

there exists $i \in \{1, 2\}$, such that the expected reward at step 1 is at most $\frac{1}{2}(r_g + r_b)$ in $\mathcal{M}_i$. Then the expected total return in $\mathcal{M}_i$ is at most $\frac{1}{2}(r_b + r_g) + (H-1)r_g$, so

$$\left| g^* - J_i(\pi^1, g^*) \right| \ge \frac{1}{2}(r_g - r_b).$$

Here $J_i(\pi^1, g^*)$ is the expected return of policy $\pi^1$ conditioning on desired RTG $g^*$ in MDP $\mathcal{M}_i$.

In particular, let $r_g = 1$ and $r_b = 0$, then we have $\left| g^* - J_i(\pi^1, g^*) \right| \ge \frac{1}{2}$. So we complete the proof.

## B.2 RCSL FAILS TO STITCH TRAJECTORIES

In this section, we give a counterexample to show that the decision transformer (Chen et al., 2021), a notable representative of RCSL, fails to stitch trajectories even when the MDP is deterministic. Even when the dataset covers all state-action pairs of this MDP, the decision transformer cannot act optimally with probability 1.

We abuse the notation of trajectory: for any $\tau$ in the dataset $\mathcal{D}$, we say its sub-trajectory $\tau[i:j]$ is a trajectory in $\mathcal{D}$, written as $\tau[i:j] \in \mathcal{D}$.

Here we make an assumption of the generalization ability of the return-conditioned policy. When faced with a trajectory not in the dataset, the policy outputs a mixture of the policy for trajectories in the dataset. For a discrete state space, we assume that the policy only use information related to each state, not ensembling different states. Thus, the used trajectories for mixture policy should have the same last state as that of the unseen trajectory. To formalize, we have Assumption 1.

**Assumption 1.** *Assume that the return-conditioned policy trained on the dataset $\mathcal{D}$ has a mapping function $f_{\mathcal{D}}$ which*

1. *maps an unseen trajectory $\tau$ to $\{(\tau_1, w_1), (\tau_2, w_2), \ldots\}$ where $\tau_i \in \mathcal{D}$, the last state of $\tau_i$ is equal to the last state of $\tau$, and $w_1, w_2, \ldots$ form a probability simplex;*
2. *maps any $\tau \in \mathcal{D}$ to $\{(\tau, 1)\}$.*

Under Assumption 1, the return-conditioned policy proceeds in the following way (Algorithm 3):

**Remark 4.** *For the weighted-mixture part of Assumption 1, we can view a new trajectory as an interpolation of several trajectories in the dataset. Thus, it is intuitive to view the output policy as an interpolation as well. For the "same last state" requirement, this is satisfied in the Markovian RCSL framework. This requirement is also without loss of generality for most generic MDPs, whose states are independent.*

---

**Algorithm 3** Generalization behavior of a return-conditioned policy

---
1: **Input:** Return-conditioned policy $\pi : \mathcal{S} \times \mathbb{R} \to \Delta(\mathcal{A})$ with a mapping function $f_{\mathcal{D}}$, desired RTG $g_1$.
2: Observe $s_1$.
3: Set $\tau \leftarrow (1, s_1, g_1)$.
4: **for** $h = 1, 2, \cdots, H$ **do**
5:     Sample action $a_h \sim \sum_{(\tau_i, w_i) \in f_{\mathcal{D}}(\tau)} w_i \pi(\cdot | \tau_i)$.
6:     Observe reward $r_h$ and next state $s_{h+1}$.
7:     Update $g_{h+1} \leftarrow g_h - r_h$ and $\tau \leftarrow (\tau, a_h; h+1, s_{h+1}, g_{h+1})$.

---

Now we are ready to prove Theorem 3.

*Proof of Theorem 3.* Construct $\mathcal{M}$ in the following way:

- Horizon: $H = 2$.

- State space: $\mathcal{S} = \{s_1, s_2, s_3\}$.

- Action space: $\mathcal{A} = \{a_1, a_2\}$.

- Initial state: The initial state is $s_1$, i.e., $\mu(s_1) = 1$.

- Transition: For any state $s_i (i \in \{1, 2\})$, any action $a_j (j \in \{1, 2\})$, we have $P(s_{i+1} | s_i, a_j) = 1$. For $s_3$, we have $P(s_3 | s_3, a_j) = 1$.

- Reward: For any state $s_i (i \in \{1, 2, 3\})$, any action $a_j (j \in \{1, 2\})$, we have $r(s_i, a_j) = j - 1$.

The illustration is shown in Figure 5.

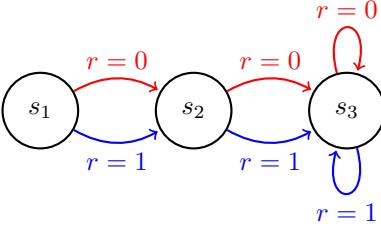

Figure 5: The counterexample showing that decision transformers cannot stitch even in a deterministic MDP. Transitions of $a_1$ are plotted in red arrows, and those of $a_2$ are in blue.

The optimal policy is to take $a_2$ at both $s_1$ and $s_2$, with a total reward of 2.

The dataset is

$$\mathcal{D} = \{(s_1, 1, a_1; s_2, 1, a_2; s_3, 0),$$
$$(s_1, 1, a_2; s_2, 0, a_1; s_3, 0)\}.$$

All the state-action pairs are covered by $\mathcal{D}$, but the optimal trajectory is not in $\mathcal{D}$. Both $(s_1, 1, a_1)$ and $(s_1, 1, a_2)$ are in $\mathcal{D}$, and they have the same weight in the loss function of decision transformer:

$$\ell_{\mathcal{D}}(\pi) = -\log \pi(a_1 | s_1, 1) - \log \pi(a_2 | s_1, 1, a_1; s_2, 1)$$
$$- \log \pi(a_2 | s_1, 1) - \log \pi(a_1 | s_1, 1, a_2; s_2, 0).$$

After minimizing the loss function, both $a_1$ and $a_2$ have non-zero probability at step 1.

When we want to achieve the optimal return-to-go 2, the trajectory is $\tau = (s_1, 2)$. The return-conditioned policy $\pi$ is fed with a mixture of trajectories in $f(\tau) = \{[(s_1, 1), 1]\}$, because $\tau \notin \mathcal{D}$. Thus, the policy has non-zero probability mass on the sub-optimal action $a_1$. $\qquad\square$

# C    ABLATION STUDIES

In this section, we study the following questions about MBRCSL:

(1) How does a low-quality dynamics model or behavior policy affect the performance of MBRCSL?

(2) What is the relationshape between MBRCSL performance and the rollout dataset size?

(3) How does the final output policy affect the performance of MBRCSL?

(4) How does data distribution in offline dataset influence the performance of MBRCSL?s

## C.1    DYNAMICS AND BEHAVIOR POLICY ARCHITECTURE

We tested on Pick and Place task to answer question (1). We designed three variants of MBRCSL:

- "Mlp-Dyn" replaces the transformer dynamics model with an MLP ensemble dynamics model.
- "Mlp-Beh" replaces the diffusion behavior policy with a MLP behavior policy.
- "Trans-Beh" replaces the diffusion behavior policy with a transformer behavior policy.

According to Table 5, all three variants fail to reach a positive success rate. The potential reason is that all of them fail to generate optimal rollout trajectories. For Mlp-Dyn, the low-quality dynamics could mislead behavior policy into wrong actions. For Mlp-Beh and Trans-Beh, both of them applies a unimodal behavior policy, which lacks the ability the stitch together segments of different trajectoreis.

Table 5: Ablation study about dynamics and behavior policy architecture on Pick and Place task. We use 4 seeds for each evaluation.

| Task | MBRCSL (ours) | Mlp-Dyn | Mlp-Beh | Trans-Beh |
|------|---------------|---------|---------|-----------|
| PickPlace | **0.48±0.04** | $0 \pm 0$ | $0 \pm 0$ | $0 \pm 0$ |

## C.2    ROLLOUT DATASET SIZE

To answer question (2), we adjusted the size of rollout dataset, which consists of rollout trajectories with return higher than the maximum return in the offline dataset. As shown in Figure 6, the performance increases significantly when rollout dataset size is less than 3000, and then converges between 0.45 and 0.5 afterwards. This is potentially because small rollout datasets are sensitive to less accurate dynamics estimate, so that inaccuarate rollouts will lower the output policy's performance.

## C.3    OUTPUT POLICY ARCHITECTURE

To answer question (3), we investigate the strength of RCSL on Point Maze task by replacing the final RCSL policy with the following architectures:

- CQL policy. That is, we train CQL on the rollout dataset, labeled as "MBCQL"
- Return-conditioned Gaussian policy (Emmons et al., 2022), labeled as "MBRCSL-Gaussian".
- Decision Transformer (Chen et al., 2021), labeled as "MBDT".
- MLP policy trained via behavior cloning on the rollout dataset, labeled as "MBBC".

As shown in Table 6, RCSL output policy achieves higher averaged return than CQL with lower variance, demonstrating again the advantage of RCSL over DP-based methods on near-expert datasets. The Gaussian policy captures less optimal behaviors due to the modeling of stochasticity. Decision transformer output policy achieves lower performance than the MLP return-conditioned policy used in MBRCSL. MBRCSL achieves higher expected return than MBBC, because RCSL extracts the policy with highest return from the rollout dataset, while BC can only learn an averaged policy in rollout dataset.

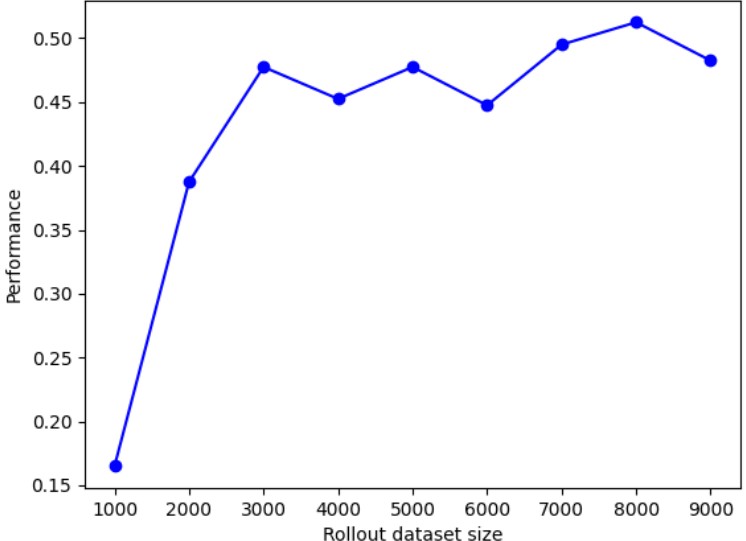

Figure 6: Ablation study about rollout dataset size on Pick and Place task. We use 4 seeds for each evaluation.

Table 6: Comparison between different output policies on Point Maze. All algorithms use rollout dataset collected by MBRCSL We use 4 seeds for evaluations.

| Task | MBRCSL (ours) | MBCQL | MBRCSL-Gaussian | MBDT | MBBC |
|---|---|---|---|---|---|
| Pointmaze | **91.5±7.1** | 49.0±24.1 | 62.7±31.4 | 79.8±4.6 | 83.0±7.5 |

### C.4  DATA DISTRIBUTION

Finally, we study the effect of data distribution on Point Maze task. We adjust the ratio between the two kinds of trajectories in Point Maze dataset (cf. Figure 3). To be concrete, let

$$\text{ratio} := \frac{\text{Number of trajectories } S \rightarrow A \rightarrow B \rightarrow G}{\text{Offline dataset size}}. \tag{5}$$

We recorded the averaged performance w.r.t. different ratio value. In addition, we investigated the effect of ratio on rollout process, so we recorded "high return rate", i.e., the ratio between rollout dataset size (number of rollout trajectories with returns larger than the maximum in offline dataset), and the total number of generated rollout trajectories. During the experiments, we fixed the rollout dataset size.

According to Figure 7, we can see that the averaged return does not show an evident correlation with ratio. This is potentially because the rollout dataset consists only of rollout trajectories whose returns are higher than the maximum offline dataset, i.e., nearly optimal. However, the high return rate increases when ratio approaches 0.5, i.e., uniform distribution. In a skewed dataset, less frequent trajectories will attract less attention to the behavior policy, so that the behavior cannot successfully learn those trajectories, leading to a drop in the ratio of optimal stitched trajectories in rollouts.

### D  BELLMAN COMPLETENESS CANNOT BE SATISFIED BY SIMPLY INCREASING MODEL SIZE

We claim that the poor performance of DP-based offline RL baselines in Section 5 is not due to small model size. For this aim, we tested CQL on Point Maze tasks with different model size. The Q-network is implemented with a 4-layer MLP network, with each hidden layer width equal to hyper-parameter "dim". The table below presents the averaged return of CQL on 4 seeds under different dim values:

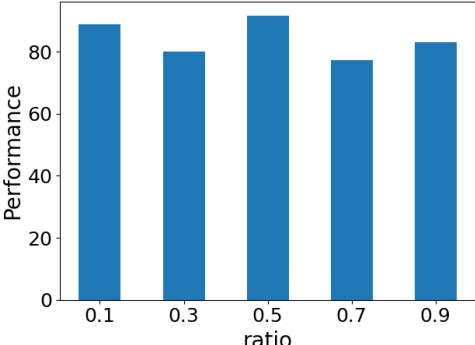 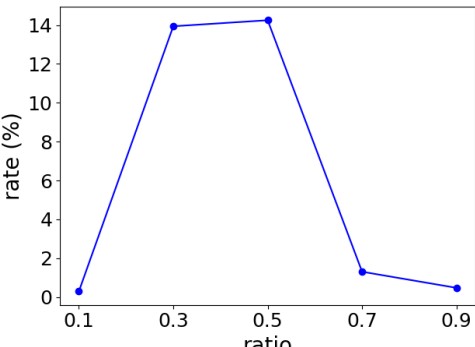

Figure 7: Ablation study about data distribution on Point Maze. The left subfigure shows the averaged return for different data distribution, and the right subfigure shows the high return rate w.r.t. different data distribution. We use 4 seeds for each evaluation.

Table 7: Influence of model size on performance of CQL on Point Maze. We use 4 seeds for each evaluation. The averaged return does not increase with model size.

| CQL model size | dim=256 | dim=512 | dim=1024 | dim=2048 |
|---|---|---|---|---|
| PointMaze averaged return | 34.8 | 52.4 | 43.7 | 44.2 |

We can see from Table 7 that the performance of CQL does not increase with a larger model. This is because Bellman completeness cannot be satisfied by simply increasing model size. In fact, it is a property about the function class on the MDP, so improving CQL would require modifying the architecture to satisfy this condition. However, it is highly non-trivial to design a function class that satisfies Bellman completeness.

## E    EXPERIMENT DETAILS

In this section, we introduce all details in our experiments. Code for reproducing the results is available at `https://github.com/zhaoyizhou1123/mbrcsl`.

### E.1    DETAILS OF POINT MAZE EXPERIMENT

**Environment.** We use the Point Maze from Gymnasium-Robotics (`https://robotics.farama.org/envs/maze/point_maze/`), which is a re-implementation of D4RL Maze2D environment (Fu et al., 2020). We choose the dense reward setting, in which the reward is the exponential of negative Euclidean distance between current ball position and goal position. We designed a customized maze and associated dataset, as illustrated in Section 5.1. We fix horizon to be 200.

**Baselines.** CQL (Kumar et al., 2020) (including CQL part in MBCQL), COMBO (Yu et al., 2021) and MOPO (Yu et al., 2020) implementations are based on OfflineRL-Kit (`https://github.com/yihaosun1124/OfflineRL-Kit`), an open-source offline RL library. We keep the default hyperparameter choices. We implemented DT (Chen et al., 2021) as well %BC based on official Decision Transformer repository (`https://github.com/kzl/decision-transformer`). We set context length to 20 and batch size to 256. We condition initial return of DT on the maximum dataset return. Other hyper-parameters are the same as Chen et al. (2021, Table 9).

**MBRCSL.** The ensemble dynamics is implemented based on OfflineRL-Kit (`https://github.com/yihaosun1124/OfflineRL-Kit`). We keep default hyperparameter choice. The diffusion behavior policy is adapted from Diffusion Policy repository (`https://github.com/real-stanford/diffusion_policy`), which is modified to a Markovian policy. We keep default hyperparameter choice, except that we use 10 diffusion steps. Empirically, this leads to good trajectory result with relatively small computational cost. Hyperparameter selections for output policy are listed in Table 8.

Table 8: Hyperparameters of output return-conditioned policy in MBRCSL for Point Maze experiments

| Hyperparameter | Value |
|---|---|
| Hidden layers | 2 |
| Layer width | 1024 |
| Nonlinearity | ReLU |
| Learning rate | 5e-5 |
| Batch size | 256 |
| Initial RTG | Maximum return in rollout dataset |
| Policy Output | Deterministic |

## E.2 DETAILS OF SIMULATED ROBOTIC EXPERIMENTS

**Environment.** The environments are implemented by Roboverse (`https://github.com/avisingh599/roboverse`). All three tasks (PickPlace, ClosedDrawer, BlockedDrawer) use the sparse-reward setting, i.e., the reward is received only at the end of an episode, which is 1 for completing all the required phases, and 0 otherwise. The action is 8-dimensional for all three tasks.

For PickPlace, the initial of position of object is random, but the position of tray, in which we need to put the object is fixed. The observation is 17-dimensional, consisting of the object's position (3 dimensions), object's orientation (4 dimensions) and the robot arm's state (10 dimensions). The horizon is fixed to 40.

For ClosedDrawer, both the top drawer and bottom drawer start out closed, and the object is placed inside the bottom drawer. The observation is 19-dimensional, consisting of object's position (3 dimensions), object's orientation (4 dimensions), the robot arm's state (10 dimensions), the x-coordinate of the top drawer (1 dimension) and the x-coordinate of the bottom drawer (1 dimension). The horizon is fixed to 50.

For BlockedDrawer, the top drawer starts out open, while the bottom drawer starts out closed, and the object is placed inside the bottom drawer. The observation space is the same as that of ClosedDrawer, i.e., 19-dimensional. The horizon is fixed to 80.

**Baselines.** CQL (Kumar et al., 2020) (including CQL part in MBCQL), COMBO (Yu et al., 2021) are based on OfflineRL-Kit (`https://github.com/yihaosun1124/OfflineRL-Kit`). We keep the default hyperparameter choices, except that we set conservative penalty $\alpha$ in both CQL and COMBO to 1.0, which leads to higher successful rate for CQL. We implemented DT (Chen et al., 2021) based on official Decision Transformer repository (`https://github.com/kzl/decision-transformer`). We set context length to 20 and batch size to 256. We condition initial return of DT on the maximum dataset return. Other hyper-parameters are the same as Chen et al. (2021, Table 9).

**MBRCSL.** The diffusion behavior policy is the same as in Point Maze Experiment, except that we use 5 diffusion steps, which has the best performance empirically. The transformer model in dynamics is based on MinGPT (`https://github.com/karpathy/minGPT`), an open-source simple re-implementation of GPT (`https://github.com/openai/gpt-2`). We adapt it into a dynamics model $\widehat{T}_\theta(s', r|s, a)$. Output policy is implemented with autoregressive model. The detailed hyperparameter choices are listed in Table 9.

## E.3 DETAILS OF LINEARQ EXPERIMENT

The algorithms are implemented with minimal extra tricks, so that they resemble their theoretical versions discussed in Section 3.1.

The policy in MLP-RCSL algorithm is implemented with a MLP model. It takes the current state and return as input, and outputs a predicted action. We train the policy to minimize the MSE error of action (cf. Eq. (2)).

The $Q$-function in $Q$-learning algorithm is implemented with a MLP model. It takes current state $s$ as input and outputs $(\widehat{Q}(s, a = 0), \widehat{Q}(s, a = 1))$, the estimated optimal $Q$-functions on both $a = 0$ and $a = 1$. During evaluation, the action with higher estimated $Q$-function value is taken. We maintain a

Table 9: Hyperparameters of MBRCSL for Pick-and-Place experiments

| Hyperparameter | Value |
|---|---|
| Horizon | 40 PickPlace |
| | 50 ClosedDrawer |
| | 80 BlockedDrawer |
| Dynamics/architecture | transformer |
| Dynamics/number of layers | 4 |
| Dynamics/number of attention heads | 4 |
| Dynamics/embedding dimension | 32 |
| Dynamics/batch size | 256 |
| Dynamics/learning rate | 1e-3 |
| Behavior/architecture | diffuser |
| Behavior/diffusion steps | 5 |
| Behavior/batch size | 256 |
| Behavior/epochs | 30 PickPlace, |
| | 40 ClosedDrawer, BlockedDrawer |
| RCSL/architecture | autoregressive |
| RCSL/hidden layers | 4 |
| RCSL/layer width | 200 |
| RCSL/nonlinearity | LeakyReLU |
| RCSL/learning rate | 1e-3 |
| RCSL/batch size | 256 |
| RCSL/Initial RTG | Maximum return in rollout dataset |

separate target $Q$-network and update target network every 10 epochs, so that the loss can converge before target network update.

We train both algorithms for 300 epochs and record their performance on the last epoch.

