# OpenReview forum: "Free from Bellman Completeness: Trajectory Stitching via Model-based Return-conditioned Supervised Learning"
_ICLR.cc/2024/Conference — ICLR 2024 poster_

### Official Review · Reviewer_BsaB · 2023-10-25

**Soundness:** 3 good
**Presentation:** 3 good
**Contribution:** 3 good
**Rating:** 6
**Confidence:** 3

**Summary:**

The paper proposes a novel method for off-line reinforcement learning based on return-conditioned supervising learning (RCSL) algorithms for policy selection working on an augmented data set of roll-out trajectories generated by means of a dynamics model learned from actually experienced trajectories. An empirical evaluation demonstrates the advantages of the proposed method on several decision problems. In addition, the paper provided theoretical analysis of why return-conditioned algorithms cannot stitch an optimal policy from segments of trajectories contained in a data set, thus motivating the idea to augment such a data set with rollouts.

**Strengths:**

The main strength of the paper is the combination of RCSL with rollouts using a learned model for the purposes of augmentation of a collected data set of trajectories. The use of RCSL allows the use of machine learning models that are only adequate for representing the optimal policy, and not necessarily for representing the optimal Q-function of the problem, too, as required by methods based on dynamic programming (Bellman back-ups). This makes the selection of the neural network models for a given decision problem easier. On the other hand, the use of rollouts produces by means of a learned model increases the set of trajectories that the RCSL will be selecting from, thus increasing the cumulative reward of the resulting policy.

**Weaknesses:**

The inability of RCSL to stitch an optimal trajectory from multiple segments of trajectories in the data set it is working with seems like a major limitation. If it is true that RCSL can learn the optimal policy only if the optimal trajectory is in the data set, then for optimality, the augmented data set must be large enough to contain this trajectory. But, it is not clear how many rollouts will be needed for this, even in some probabilistic sense.

Another weakness of the paper is that it is hard to follow the proposed methods. For example, the RCSL method can be explained better.

A minor typo, in the abstract: "unde" -> "under"

**Questions:**

How exactly is the policy represented in the comparison in Section 3.1? To begin with, there seems to be a discrepancy between the kind of policies introduced in Section 2, where a policy maps trajectories to probability distributions over the action set, and the one in Section 3.1, where the policy is deterministic, and outputs a single action. Equation 2 is compatible only with the latter type of policy, and only if the actions are real-valued. But, the projection f(a) seems to be a thresholding operation, returning only possible values of 0 and 1. They could be used as real-valued actions, but then, how are problems with more than two actions represented? Also, if f(a) is a hard thresholding, it is not differentiable, how is the minimization problem in the MLP-RCSL solved?

Also, are the indices for tau[i, j] in the first equation in Section 2.2 correct? For cx=1, we get i>j, which seems to contradict the definition of a sub-sequence tau[i, j]. Also, isn't s_h already part of that sequence, why is it listed as a separate argument of the policy?

---

> ### Author Response · Authors · 2023-11-20
> **Response to Reviewer BsaB**
>
> Thank you for your constructive feedback. We address your comments below.
>
> >  **Weakness 1. For optimality, the augmented data set must be large enough to contain this trajectory. But, it is not clear how many rollouts will be needed for this, even in some probabilistic sense.**
>
> We tested the ratio between rollout dataset size (number of rollout trajectories with returns larger than the maximum in offline dataset), and the total number of generated rollout trajectories. The result is shown in the table below:
>
> | Tasks | Pointmaze | PickPlace | ClosedDrawer | BlockedDrawer |
> | ----- | --------- | --------- | ------------ | ------------- |
> | ratio | 14.3%     | 19.2%     | 25.7%        | 63.2%         |
>
> The results are averaged on 4 seeds. We can see that the probability is acceptable in our benchmarks considered. It is true that the probability will reduce if the optimal trajectory consists of more trajectory segments in the offline dataset. We will work on this problem as future work.
>
>
>
> > **Weakness 2.  It is hard to follow the proposed methods. For example, the RCSL method can be explained better.**
>
> MBRCSL works as follows: First, we learn an approximate dynamics model as well an expressive behavior policy from the offline dataset. Then we rollout the behavior policy on the dynamics model to generate synthetic trajectories, which stitch together trajectory segments in the offline dataset. In this process, potentially optimal trajectories will be generated, although the averaged return of behavior policy could be low.  We aggregate potentially optimal trajectories into a new rollout dataset by picking out trajectories with estimated returns larger than the maximum return in the dataset. Finally, we apply RCSL on the rollout dataset, to further extract the optimal policy.
>
> RCSL learns a distribution of actions in each state conditioned on a desired return-to-go by directly applying supervised learning on the trajectories in the dataset. RCSL is a standard method studied in [1, 2]. We formalize RCSL framework in Section 2.2 for rigorous theoretical analysis.
>
> [1] David Brandfonbrener, Alberto Bietti, Jacob Buckman, Romain Laroche, and Joan Bruna. When does return-conditioned supervised learning work for offline reinforcement learning? Advances in Neural Information Processing Systems, 35:1542–1553, 2022.
> [2] Scott Emmons, Benjamin Eysenbach, Ilya Kostrikov, and Sergey Levine. Rvs: What is essential for offline rl via supervised learning?, 2022.
>
> > **Question 1. How exactly is the policy represented in the comparison in Section 3.1?**
>
> The policy defined in Section 2. is the most general version, which is a stochastic policy. In Section 3.1, we focus on deterministic policies for RCSL. This is without loss of generality, as deterministic policy class includes the optimal policy. Equation (2) is the training objective for MLP-RCSL, which is indeed compatible to deterministic policies only.
>
> In Section 3.1, the MLP-RCSL architecture is exactly the same as described in Section 2.2, and it is trained by minimizing Equation (2). The projection $f(a)$ is a component external to MLP-RCSL. It is added to MLP-RCSL to ensure that the output action lies exactly in the action space ($\{0,1\}$) of the example considered in Theorem 1 (cf. Figure 2). For other MDPs, the projection can be omitted or adapted to the action space. We have revised the formulation in Section 3.1.
>
> > **Question 2. Are the indices for $\tau$[i: j] in the first equation in Section 2.2 correct?**
>
> The indices are correct according to our definition in Section 2, paragraph "Trajectories", in which $\tau[i:j] = (s_i,g_i,a_i,\cdots,s_j,g_j,a_j)$. When cx=1, the term $\tau[h-cx+1:h-1 ]$ in $\pi^{\text{cx}}(a_h|\tau[h-\text{cx}+1: h-1],s_h,g_h)$ is omitted and the resulting policy is $\pi^1 (a_h|s_h,g_h)$. We have added a clarification in Section 2.2.
>
>
> > **Weakness 3. A minor typo in the abstract.**
>
> Thanks for pointing out. We have fixed it in our paper.

---

> > ### Comment · Reviewer_BsaB · 2023-11-21
> > **Response apprecaited**
> >
> > I am satisfied with the way the authors addressed my concerns and questions, and will raise the evaluation by one step.

---

### Official Review · Reviewer_9pmM · 2023-10-27

**Soundness:** 3 good
**Presentation:** 4 excellent
**Contribution:** 3 good
**Rating:** 8
**Confidence:** 4

**Summary:**

This paper presents MBRCSL, a return-conditioned supervised learning algorithm that improves on one of the major limitations of that algorithm category, the inability to perform trajectory stitching, by adding model-based rollouts that synthesize multiple trajectories to augment the training data. The authors also prove that under certain assumptions RCSL algorithms converge with weaker assumptions about the contents of the offline dataset compared to offline Q-learning methods.

**Strengths:**

I found this paper very interesting- it combines non-trivial theoretical and experimental arguments in a way I don't see often, and does so in a very comprehensible way, speaking as someone who hasn't worked on offline RL and isn't deeply familiar with the literature on the topic.

In general I liked the ideas presented- they rely on some assumptions about the dataset and MDP(s) of interest, but these are mostly stated up front and seem if not trivial in practice at least a reasonable starting point for future work to relax to some degree. The point about realizability versus Bellman completeness seems well put to me.

**Weaknesses:**

My biggest criticism of this paper is that it doesn't really have any ablations or demonstrate the experimental limitations of MBRCSL at all. The theoretical assumptions are all clearly stated, but it's not clear how far these will stretch in practice.

I think what's already here is clearly argued and significant, so I'm inclined to recommend acceptance regardless, but some more stress tests to see where assumptions break down would greatly improve the paper. I've added some questions below in that vein, hopefully they can be useful for either this paper or future work building on it.

**Questions:**

-For the two tasks tested on, the poor performance of CQL and co. is claimed to be due to issues with Bellman completeness. Could it be reasoned about or experimentally demonstrated what additional data is needed to bring those algorithms up to performance levels similar to MBRCSL? I imagine the additions needed are non-trivial, which would be a mark in MBRCSL's favor, but I'm curious to know how hard it is to satisfy the requirements of offline Q-learning algorithms in comparison. Basically, how hard of a constraint to satisfy is the need for Bellman completeness in practice versus theory?

-I'd be interested to see a non-tabular environment testing how model size/parameter count is affected by RCSL versus Q-learning. The theoretical argument here seems strong, so I'm curious if performance as a function of model size ends up being meaningfully different between RCSL methods and Q-learning methods.

-MBRCSL relies on a good forward dynamics model and a behavior policy to allow trajectory stitching, and while this is a fair requirement I'm curious what happens when those two components are lower in quality (for example, a weaker dynamics model versus a stronger one). Does performance degrade smoothly, or is a high-quality model essential?

-In the same vein, what happens if the rollout budget is reduced? How does MBRCSL degrade with limited synthetic data?

-What happens to MBRCSL if dataset coverage is non-uniform? I can imagine something like a constructed dataset that has a tunable degree of biased-ness in coverage being informative here.

---

> ### Author Response · Authors · 2023-11-20
> **Response to Reviewer 9pmM**
>
> Thank you for your positive feedback and for finding our paper “interesting” and "comprehensive." We address your comments below.
>
> > **Question 1. For the two tasks tested on, the poor performance of CQL and co. is claimed to be due to issues with Bellman completeness. Could it be reasoned about or experimentally demonstrated what additional data is needed to bring those algorithms up to performance levels similar to MBRCSL?**
>
> We tested CQL on Point Maze tasks with different model size. The Q-network is implemented with a 4-layer MLP network, with each hidden layer width equal to hyper-parameter `dim`. The table below presents the averaged return of CQL on 4 seeds under different `dim` values:
>
> | Task      | dim=256 | dim=512 | dim=1024 | dim=2048 |
> | --------- | ------- | ------- | -------- | -------- |
> | PointMaze | 34.8    | 52.4    | 43.7     | 44.2     |
>
> We can see that the performance of CQL does not increase with a larger model. This is because Bellman completeness cannot be satisfied by simply increasing model size. In fact, it is a property about the function class on the MDP, so improving CQL would require modifying the architecture to satisfy this condition. However, it is highly non-trivial to design a function class that satisfies Bellman completeness. We added this discussion in Appendix D of our paper.
>
> > **Question 2. A non-tabular environment testing how model size/parameter count is affected by RCSL versus Q-learning**
>
> We compared RCSL and CQL on Point Maze task with expert dataset (referred to as "Pointmaze expert"). To be specific, the dataset contains two kinds of trajectories: $S\to A \to B$ and $S\to M \to G$ (optimal trajectory), and the highest return in dataset is 137.9. Please refer to Figure 3 of Section 5 for the maze illustration. Both RCSL policy and CQL Q-network are implemented with three-layer MLP networks, with every hidden layer width equal to hyper-parameter `dim`. We recorded the averaged returns on 4 seeds under different choices of `dim`:
>
>
> |      | dim=64 | dim=128 | dim=256 | dim=512 | dim=1024 |
> | ---- | ------ | ------- | ------- | ------- | -------- |
> | RCSL | 89.6   | 97.8    | 93.3    | 113     | 105      |
> | CQL  | 11.5   | 11.7    | 14.7    | 37.8    | 23       |
>
> CQL performance is much worse than RCSL on all model size, though both algorithms tend to achieve a higher averaged return when model size increases (especially for `dim`$\leq 512$). This result supports the claim in Section 3.1 that RCSL can outperform DP-based offline RL methods, as RCSL avoids Bellman completeness requirements. We added this experiment in Appendix A.3 of the paper.
>
>
>
> > **Question 3. MBRCSL relies on a good forward dynamics model and a behavior policy to allow trajectory stitching, and while this is a fair requirement I'm curious what happens when those two components are lower in quality (for example, a weaker dynamics model versus a stronger one). Does performance degrade smoothly, or is a high-quality model essential?**
>
> We have added an ablation study to study the dynamics model and behavior policy, please refer to Appendix C.1. We designed three variants of MBRCSL:
>
> - “Mlp-Dyn” replaces the transformer dynamics model with an MLP ensemble dynamics model.
> - “Mlp-Beh” replaces the diffusion behavior policy with a MLP behavior policy.
> - “Trans-Beh” replaces the diffusion behavior policy with a transformer behavior policy.
>
> We tested on PickPlace environment and got the following results:
>
> | Task      | MBRCSL (ours) | Mlp-Dyn | Mlp-Beh | Trans-Beh |
> | --------- | ------------- | ------- | ------- | --------- |
> | PickPlace | 0.48±0.04     | 0 ± 0   | 0 ± 0   | 0 ± 0     |
>
> We can see that both the quality of dynamics and that of behavior policy are crucial to final performance. One reason for failure of the three variants is that none of them are able to generate optimal trajectories. For Mlp-Dyn, the low-quality dynamics will mislead behavior policy into wrong actions. For Mlp-Beh and Trans-Beh, both of them apply the unimodal behavior policy, which lacks the ability the stitch together segments of different trajectories.

---

> ### Author Response · Authors · 2023-11-20
> **Response to Reviewer 9pmM (2)**
>
> > **Question 4.  What happens if the rollout budget is reduced?**
>
> We added an ablation study in our paper, please refer to Appendix C.2. As shown in the table below, we adjusted rollout dataset size (numbers in the first row) and recorded the averaged success rate (numbers in the second row) on PickPlace task.
>
> | Task      | 1000   | 2000   | 3000   | 4000   | 5000   | 6000   | 7000   | 8000   | 9000   |
> | --------- | ------ | ------ | ------ | ------ | ------ | ------ | ------ | ------ | ------ |
> | PickPlace | 0.1650 | 0.3875 | 0.4775 | 0.4525 | 0.4775 | 0.4475 | 0.4950 | 0.5125 | 0.4825 |
>
> We can see that the performance increases significantly when rollout dataset size is less than 3000. One reason is that small rollout datasets are sensitive to less accurate dynamics estimate, so that inaccuarate rollouts will lower the output policy’s performance. The performance then converges between 0.45 and 0.52 when rollout dataset size is large than 3000.
>
>
>
> >  **Question 5. What happens to MBRCSL if dataset coverage is non-uniform?**
>
> We tested the effect of data distribution on Point Maze. We fixed the offline dataset size and adjusted the ratio between the two kinds of trajectories in Point Maze dataset (cf. Figure 3). To be concrete, let $\text{ratio} := \text{Number of trajectories S → A → B → G} / \text{ Offline dataset size}$. We recorded the averaged performance w.r.t. different ratio value. In addition, we investigated the effect of ratio on rollout process, so we recorded "*high return rate*", i.e., the ratio between rollout dataset size (number of rollout trajectories with returns larger than the maximum in offline dataset), and the total number of generated rollout trajectories. During the experiments, we fixed the rollout dataset size. The results are shown below (All results are averaged on 4 seeds):
>
> | ratio            | 0.1   | 0.3    | 0.5    | 0.7   | 0.9   |
> | ---------------- | ----- | ------ | ------ | ----- | ----- |
> | return           | 88.9  | 79.9   | 91.5   | 77.2  | 82.9  |
> | high return rate | 0.29% | 13.94% | 14.25% | 1.30% | 0.47% |
>
> We can see that the averaged return does not show an evident correlation with ratio. This is potentially because the rollout dataset consists only of rollout trajectories whose returns are higher than the maximum offline dataset, i.e., nearly optimal. However, the high return rate increases when ratio approaches 0.5, i.e., uniform distribution. In a skewed dataset, less frequent trajectories will attract less attention to the behavior policy, so that the behavior cannot successfully learn those trajectories, leading to a drop in the ratio of optimal stitched trajectories in rollouts.
>
> We added these results as an ablation study in our paper, please refer to Appendix C.4.

---

> ### Comment · Reviewer_9pmM · 2023-11-21
> **Response to rebuttal**
>
> Thanks for the additional explanation and experimental validation! These additional tests give me more confidence in the experimental results, and as such I've raised my score. I think with these additions the paper is in a good place now.

---

### Official Review · Reviewer_HXBB · 2023-10-31

**Soundness:** 3 good
**Presentation:** 3 good
**Contribution:** 3 good
**Rating:** 5
**Confidence:** 5

**Summary:**

The paper tackles a central challenge in offline reinforcement learning - the difficulty of stitching together suboptimal trajectories into optimal behavior, while using function approximation. Standard off-policy algorithms like Q-learning rely on dynamic programming and the strong requirement of Bellman completeness, which can be problematic to satisfy with function approximation.

In contrast, the paper shows return-conditioned supervised learning (RCSL) methods that directly train policies on trajectories need only the easier-to-satisfy realizability condition. However, RCSL methods cannot stitch trajectories. To get the best of both worlds, the paper proposes model-based RCSL (MBRCSL), creatively combining RCSL and model-based RL. MBRCSL trains a dynamics model, uses it to sample high-performing trajectories, and trains an RCSL policy on these optimal trajectories. This enables trajectory stitching without requiring Bellman completeness.

Theoretically, the paper sharply characterizes when RCSL can outperform Q-learning in simple settings and why RCSL fails at stitching. Empirically, MBRCSL demonstrates clear improvements over model-free and model-based baselines on Point Maze and Pick-and-Place environments.

**Strengths:**

Provides concrete theoretical characterization showing RCSL only requires realizability while Q-learning requires stronger Bellman completeness.

Proves fundamental limitation that Markovian RCSL policies cannot perform trajectory stitching.

Interesting idea of using model rollouts to generate near-optimal trajectories then training RCSL policy.

Modular framework allows flexible choice of model architecture.

Good empirical results on Point Maze and Pick-and-Place. Outperforms model-free baselines.

**Weaknesses:**

Limited experimental validation on only 2 environments. More complex domains needed.

No comparison to other state-of-the-art model-based offline RL algorithms

Theoretical results consider tabular case. Unclear if insights extend directly to function approximation.

**Questions:**

How does MBRCSL compare to other state-of-the-art offline RL algorithm on benchmark tasks?

How does MBRCSL compare to prior model-based offline RL methods like MOReL, MOPO, COMBO including sample complexity and compute? E.g. your approach requires accurate learned dynamics model and uncertainty modeling is less sophisticated than ensemble approaches like MOPO. No explicit handling of OOD actions unlike conservative methods means the algorithm could be brittle on highly suboptimal datasets- could you comment on this?

Can you formally characterize conditions under which model rollouts provide sufficient coverage for optimality of RCSL?

Can you incorporate offline constraints into RCSL policy learning?

Can you analyze sample efficiency of MBRCSL with limited suboptimal data?

How many trajectories do you need before performing BC? I would imagine the performance of the algorithms depends a lot on this?

In general, I'm quite surprised the algorithms works as explained because if BC is a policy that combines the best and worst trajectories in the dataset I don't understand how it can find trajectories that are better than the best ones. Unfortunately the experiments aren't particularly convincing in this respect.




Is end-to-end training of model and RCSL policy possible?

What are the most important architectural choices for dynamics model and RCSL policy?

Can MBRCSL be extended to stochastic environments?

The theoretical results comparing RCSL and Q-learning are derived in the tabular setting. However, function approximation is crucial for offline RL in complex environments with large state spaces. Can you discuss how the theoretical insights around Bellman completeness and realizability could be extended to the function approximation setting? Analyzing the hidden layer width requirements for MLP-RCSL versus MLP Q-learning could provide useful insights into when RCSL methods might have advantages over dynamic programming with function approximation for offline RL.

**Details Of Ethics Concerns:**

No concers.

---

> ### Author Response · Authors · 2023-11-20
> **Response to Reviewer HXBB**
>
> Thank you for your careful review and constructive feedback. We address your questions below.
>
> > **Weakness 1. Limited experimental validation on only 2 environments.**
>
> We added two simulated robotics tasks, "ClosedDrawer" and "BlockedDrawer". The former requires the robot to accomplish (1) Open a drawer; (2) Grasp the object in the drawer. The latter requries the robot to do (1) Close a drawer on top of the target drawer, and then open the target drawer; (2) Grasp the object in the target drawer.
>
> The results are shown below. We also added the result to our paper in Table 2 in Section 5.
>
> | Task          | MBRCSL (ours) | CQL       | COMBO | DT   | BC        |
> | ------------- | ------------- | --------- | ----- | ---- | --------- |
> | ClosedDrawer  | **0.51±0.12** | 0.11±0.08 | 0±0   | 0±0  | 0.38±0.02 |
> | BlockedDrawer | **0.68±0.09** | 0.34±0.23 | 0±0   | 0±0  | 0.61±0.02 |
>
>
>
> > **Weakness 2. No comparison to other state-of-the-art model-based offline RL algorithms.**
>
> The original submission featured comparisons to a few state-of-the-art model-based offline RL baselines (COMBO & MOPO for Point Maze, COMBO for PickPlace.) We additionally compare to MOReL (Kidambi et al., 2020) on Point Maze in Table 1 of our revised submission, and found MBRCSL to outperform MOReL by a significant margin.
>
>
>
> >  **Weakness 3. Theoretical results consider tabular case. Unclear if insights extend directly to function approximation.**
> >  **Question 11. The theoretical results comparing RCSL and Q-learning are derived in the tabular setting.**
>
> Although the class of MDPs used in Theorem 1 have finite state space, **we indeed considered function approximation in our analysis**. In particular, the realizability and Bellman completeness properties are discussed in the function approximation setting. Our theoretical analyses in Section 3 were also carried out under function approximation. In Theorem 1, the RCSL policy adopts an MLP network architecture instead of tabular representation, and $Q$-Learning also represents the $Q$-function with an MLP network.
>
> In Theorem 2, we laid no assumptions on the RCSL policy architecture, i.e., it can take any form of function approximation. In Theorem 3, we consider Decision Transformer as RCSL policy architecture.
>
> In addition, the example used in Theorem 1 can be adapted to an MDP with continuous state/action space by adding a continuous dummy variable. In general, consider an MDP $\mathcal M = (H, \mathcal S, \mathcal A, \mu, \mathcal T , r)$, where state space $\mathcal S$ and action space $\mathcal A$ are all discrete . We can construct a new MDP $\mathcal M' = (H, \mathcal S', \mathcal A', \mu', \mathcal T', r')$ with state space $\mathcal S' = \mathcal S \times \mathbb [0,1]$ and action space $\mathcal A' = \mathcal A \times \mathbb [0,1]$. The initial state distribution PDF $\mu'(s,x) = \mu(s)$, for any $(s,x)\in \mathcal S'$. The transition dynamics is $\mathcal T'(s',x'| s,x,a,y) = \mathcal T(s'|s,a)$, for any $(s',x'), (s,x)\in \mathcal S', (a,y)\in \mathcal A'$. The reward function is $r'(s,x,a,y) = r(s,a)$, for any $(s,x)\in \mathcal S', (a,y)\in \mathcal A'$. In fact, $\mathcal M'$ is equivalent to $\mathcal M$, so Theorem 1 holds on $\mathcal M'$ if and only if it holds on $\mathcal M$. This means that Theorem 1 also applies to MDPs with continuous state and action space.
>
> > **Question 1. How does MBRCSL compare to other state-of-the-art offline RL algorithm on benchmark tasks?**
>
> In Tables 1 and 2 of our original submission, we compared MBRCSL with state-of-the-art offline RL algorithms, including model-based approaches e.g. COMBO, model-free approaches e.g. CQL and RCSL methods e.g. DT. We additionally compare to MOReL on D4RL pointmaze in our revised submission. We found MBRCSL to outperform all of the baselines in our experiments.
>
>
>
> > **Question 2. How does MBRCSL compare to prior model-based offline RL methods like MOReL, MOPO, COMBO including sample complexity and compute? E.g. your approach requires accurate learned dynamics model and uncertainty modeling is less sophisticated than ensemble approaches like MOPO. No explicit handling of OOD actions unlike conservative methods means the algorithm could be brittle on highly suboptimal datasets- could you comment on this?**
>
> Offline RL methods based on dynamic programming approaches, e.g., MOReL, MOPO and COMBO requires conservatism or pessimism to avoid over-estimation of $Q$-functions for OOD actions. In contrast, MBRCSL does not rely on dynamic programming and thus does not need to handle OOD actions. In fact, the behavior policy trained via supervised learning will only take in-distribution actions, so the dynamics model can give a good estimation of states and rewards without additional conservatism. According to Table 1 of Section 5, MBRCSL indeed outperforms DP-based offline RL methods empirically.

---

> ### Author Response · Authors · 2023-11-20
> **Response to Reviewer HXBB (2)**
>
> >  **Question 3. Can you formally characterize conditions under which model rollouts provide sufficient coverage for optimality of RCSL?**
>
> If the dynamics is accurate enough and the behavior policy is multi-modal, then model rollouts would be able to generate optimal  stitched trajectories with certain probability.
> Specifically, if we obtain a uniform convergence of our estimated model for policy evaluation in the reward-free setting (which guarantees for any reward function, we can obtain an optimal policy) [1]. Then we can choose rewrds to be the indicator functions for each state, and the induced policies will give us sufficient coverage.
>
> [1] Ming Yin, Yu-Xiang Wang. Optimal Uniform OPE and Model-based Offline Reinforcement Learning in Time-Homogeneous, Reward-Free and Task-Agnostic Settings. Advances in Neural Information Processing Systems, 2021.
>
>
> >  **Question 4. Can you incorporate offline constraints into RCSL policy learning?**
>
> RCSL does not need explicit offline RL constraints like most dynamic programming based offline RL methods like CQL, COMBO and MOPO. This is because RCSL is inherently weighted supervised learning, which does not need to handle OOD actions.
>
> >  **Question 5. Can you analyze sample efficiency of MBRCSL with limited suboptimal data?**
>
> Thanks for asking. We believe it is possible to combine the analysis of  model-based offline RL [1] and recent development for RCSL [2]. We leave it as a future work to give a formal sample complexity theorem of MBRCSL.
>
> [1] Ming Yin, Yu-Xiang Wang. Optimal Uniform OPE and Model-based Offline Reinforcement Learning in Time-Homogeneous, Reward-Free and Task-Agnostic Settings. Advances in Neural Information Processing Systems, 2021.
>
> [2] David Brandfonbrener, Alberto Bietti, Jacob Buckman, Romain Laroche, and Joan Bruna. When does return-conditioned supervised learning work for offline reinforcement learning? Advances in Neural Information Processing Systems, 022
>
>
> > **Question 6. How many trajectories do you need before performing BC? I would imagine the performance of the algorithms depends a lot on this?**
>
> We added an ablation study on PickPlace task, as shown in the table below. We adjusted rollout dataset size (numbers in the first row) and recorded the averaged success rate (numbers in the second row).
>
> | Task      | 1000   | 2000   | 3000   | 4000   | 5000   | 6000   | 7000   | 8000   | 9000   |
> | --------- | ------ | ------ | ------ | ------ | ------ | ------ | ------ | ------ | ------ |
> | PickPlace | 0.1650 | 0.3875 | 0.4775 | 0.4525 | 0.4775 | 0.4475 | 0.4950 | 0.5125 | 0.4825 |
>
> We can see that the performance increases significantly when rollout dataset size is less than 3000, and then converges between 0.45 and 0.52 afterwards. We have added this ablation study to our paper in Appendix C.
>
>
>
> >  **Question 7. In general, I'm quite surprised the algorithms works as explained because if BC is a policy that combines the best and worst trajectories in the dataset I don't understand how it can find trajectories that are better than the best ones.**
>
> The behavior policy does not discern the quality of generated rollout trajectories. Instead, it only needs to stitch trajectory segments in offline dataset. As an illustration, suppose an MDP has 5 states $\mathcal S=\{A,B,C,D,E\}$, and the dataset consists of two kinds of trajectories: $A\to C \to E$ and $B\to C \to D$. Then by rolling out the behavior policy, we obtain 4 possible kinds of rollout trajectories: $A\to C \to E$, $A\to C \to D$, $B\to C \to D$, $B \to C \to E$. Some of these trajectories have high returns, and others have low returns. Given heterogeneous rollout trajectories, the output policy is able to extract the best trajectories due to return-conditioning [1-3]. To summarize, the behavior policy only provides stitched rollout trajectories with varied performance, and the return-conditioned output policy extracts the best policy from rollout dataset.
>
> [1] David Brandfonbrener, Alberto Bietti, Jacob Buckman, Romain Laroche, and Joan Bruna. When does return-conditioned supervised learning work for offline reinforcement learning? Advances in Neural Information Processing Systems, 35:1542–1553, 2022.
>
> [2] Scott Emmons, Benjamin Eysenbach, Ilya Kostrikov, and Sergey Levine. Rvs: What is essential for offline rl via supervised learning?, 2022.
>
> [3] Lili Chen, Kevin Lu, Aravind Rajeswaran, Kimin Lee, Aditya Grover, Misha Laskin, Pieter Abbeel, Aravind Srinivas, and Igor Mordatch. Decision transformer: Reinforcement learning via sequence modeling. Advances in neural information processing systems, 34:15084–15097, 2021.

---

> ### Author Response · Authors · 2023-11-20
> **Response to Reviewer HXBB (3)**
>
> >  **Question 8. Is end-to-end training of model and RCSL policy possible?**
>
> An end-to-end training scheme is not feasible in our setting. As illustrated in Figure 1, the training order is: dynamics model & behavior policy $\to$ trajectory rollout $\to$ return-conditioned output policy. This is because training of return-conditioned output policy depends on rollout data, which is generated through dynamics model and behavior policy.
>
>
>
> >  **Question 9. What are the most important architectural choices for dynamics model and RCSL policy?**
>
> The dynamics model must be accurate enough, so we recommend using a transformer model, especially in complex tasks such as the simulated robotic tasks in Section 5.2.  On the other hand, the RCSL policy architecture could be task dependent. Please refer to [1] for details.
>
> We've also added an ablation study to investigate different architectures of dynamics and RCSL policy in our paper. Please refer to Appendix C.1 and C.3.
>
> [1] Yang, S., Nachum, O., Du, Y., Wei, J., Abbeel, P., & Schuurmans, D. (2023). Foundation models for decision making: Problems, methods, and opportunities. arXiv preprint arXiv:2303.04129.
>
> >  **Question 10. Can MBRCSL be extended to stochastic environments?**
>
> As written in our conclusion, we will consider MBRCSL in stochastic environments as a direct future work. One possible solution is to learn a more general trajectory representation other than return-to-go, as described in [1, 2].
>
> [1] Paster, Keiran, Sheila McIlraith, and Jimmy Ba. "You can’t count on luck: Why decision transformers and rvs fail in stochastic environments." Advances in Neural Information Processing Systems* 35 (2022): 38966-38979.
>
> [2] Yang, M., Schuurmans, D., Abbeel, P., & Nachum, O. (2022). Dichotomy of control: Separating what you can control from what you cannot. Preprint arXiv:2210.13435.

---

> ### Author Response · Authors · 2023-11-21
> **Follow-Up**
>
> Dear Reviewer,
>
> Thank you for your time and efforts in reviewing our work. We have provided detailed clarification to address the issues raised in your comments. If our response has addressed your concerns, we would be grateful if you could re-evaluate our work.
>
> If you have any additional questions or comments, we would be happy to have further discussions.
>
> Thanks,
>
> The authors

---

### Official Review · Reviewer_YXKR · 2023-11-01

**Soundness:** 2 fair
**Presentation:** 3 good
**Contribution:** 2 fair
**Rating:** 5
**Confidence:** 3

**Summary:**

This paper aims to investigate how to make the RCSL method work in the off-policy RL setting. The paper first shows that RCSL can achieve better performance than DP-based algorithms in near deterministic environments, as long as the dataset contain optimal trajectories, and points out that this is due to that there is no need of Bellman-completeness for RCSL. This indicates that this type of methods should work if sufficient data coverage is obtained. Hence the authors propose a model-based RCSL (MBRCSL) algorithm which is able stitch together sub-optimal data while avoiding the Bellman-completeness requirements. Empriical results on D4RL benchmark also verify its effectiveness.

**Strengths:**

While previous method work (Brandfonbrener et al., 2022) gives negative theoretical results for RCSL, this paper investigates how to make it work in practice in despite of those disadvantages, while preserving its advantages. The paper is well written, easy to read and demonstrates that the proposed method may significantly outperforms SOTA offline RL methods under certain environments.

**Weaknesses:**

The proposal to address the 'trajectory stitching' issue seems idealistic - how can you ensure that sufficient data coverage can be obtained simply by rolling out a learned dynamics model?

**Questions:**

1. Theorem 1 only shows that there exists some MDPs that don't need the Bellman-completeness assumption for RCSL to work, but how to characterize such MDPs? How about other environments that lacks such characteristic?

2. In section.4, it's said that 'in many cases, this generated data provides coverge over potentially optimal trajectory and return data....', in my opinion, the policy learned by behavior cloning from suboptimal offline trajectories is highly likely to be a suboptimal policy as well, even though the dynamic model is leanred accurately, would you please make it more clear that why the trajectories sampled by the behavior policy with approximat dynamic model will become optimal?  In particular, even if the training rollouts consist of those synthetic trajectories with a return greater than the ones in offline dataset, these better rollouts are based on the approximate dynamic model and cannot guarantee the optimal trajectory. If so, how can we learn the optimal policy using the RCSL algorithm?

3. Has the authors tried MB+DT or MB+BC, i.e., replacing the RCSL component in MBRCSL with other methods without the requirement of Bellman-completeness?

4. on p.4., below section 3.2, the paper says that "RCSL cannot perform trajecatory stitching", would you please make it more clear? In my opinion, RCSL works by supervised learning, in which  part of its data features (future return or goals for each state) are generated by trajactory stitching by hand.

---

> ### Author Response · Authors · 2023-11-20
> **Response to Reviewer YXKR**
>
> Thank you for your review. We address your questions below.
>
> > **Question 1. Theorem 1 only shows that there exists some MDPs that don't need the Bellman-completeness assumption for RCSL to work, but how to characterize such MDPs? How about other environments that lacks such characteristic?**
>
> For any MDP, Bellman completeness is a **property of the function approximation class** (cf. Definition 2). For any MDP, the tabular function class always satisfies Bellman completeness, as it encompasses all possible $Q$-functions. It is easier for a function class to satisfy the realizablity requirement of RCSL than the Bellman completeness requirement of $Q$-learning because the realizabity approximation error ($\max_{Q \in \mathcal{F}}\|Q-Q^*\|$) decreases monotonically as the function class size increases, but this is not true for Bellman completeness approximation error. Theorem 1 proves this belief rigorously with a concrete example.
>
> > **Question 2. Would you please make it more clear that why the trajectories sampled by the behavior policy with approximat dynamic model will become optimal?**
> > **Weakness. How can you ensure that sufficient data coverage can be obtained simply by rolling out a learned dynamics model?**
>
> Although the behavior policy learned from offline dataset is sub-optimal in terms of expected return, it is able to stitch trajectory segments in offline dataset. As an illustration, suppose an MDP has 5 states $\mathcal S=\{A,B,C,D,E\}$. The dataset consists of two kinds of trajectories: $A\to C \to E$ and $B\to C \to D$, and the optimal trajectory is $A\to C\to D$.  Then there will be 4 possible kinds of rollout trajectories: $A\to C \to E$, $A\to C \to D$, $B\to C \to D$, $B \to C \to E$.  Therefore, the behavior policy is able to generate the optimal trajectory, albeit with low probability.  The return-conditioned output policy to find out the best trajectories in rollout dataset, by applying return-conditioned supervised learning.
>
> In fact, we tested the ratio between rollout dataset size (number of rollout trajectories with returns larger than the maximum in offline dataset), and the total number of generated rollout trajectories, as shown in the table below:
>
> | Tasks | Pointmaze | PickPlace | ClosedDrawer | BlockedDrawer |
> | ----- | --------- | --------- | ------------ | ------------- |
> | ratio | 14.3%     | 19.2%     | 25.7%        | 63.2%         |
>
> We can see that i.i.d. rollout through dynamics model indeed generates adequate high-return trajectories empirically.
>
> While the returns of rollout trajectories are generated by the learned dynamics model, the estimation error can be reduced to an acceptable rate by learning an accurate dynamics. We provide an ablation study on dynamics model architecture in Appendix C.1.
>
>
> > **Question 3. Has the authors tried MB+DT or MB+BC, i.e., replacing the RCSL component in MBRCSL with other methods without the requirement of Bellman-completeness?**
>
> We added the two experiments on Point Maze environment, as shown in the table below. We have updated this result in our paper, please refer to Appendix C.3.
>
> | Task      | MBRCSL (ours) | MBDT     | MBBC     |
> | --------- | ------------- | -------- | -------- |
> | Pointmaze | **91.5±7.1**  | 79.8±4.6 | 83.0±7.5 |
>
> We can see that Decision Transformer output policy achieves lower performance than the MLP return-conditioned policy used in MBRCSL. MBRCSL achieves higher expected return than MBBC, because RCSL extracts the policy with highest return from the rollout dataset, while BC can only learn an averaged policy in rollout dataset.

---

> ### Author Response · Authors · 2023-11-20
> **Response to Reviewer YXKR (2)**
>
> > **Question 4. In my opinion, RCSL works by supervised learning, in which part of its data features (future return or goals for each state) are generated by trajectory stitching by hand.**
>
> The return-to-gos in standard RCSL are Monte-Carlo, in that they are generated from a single trajectory in the offline dataset. On the other hand, trajctory stitching requires combining multiple trajectories. During training, the return-to-go of a trajectory $\tau$ at timestep $h$ is computed by accumulating the rewards achieved at timesteps $h'\geq h$ (cf. Section 2, paragraph "Trajectories"). During evaluation (Algorithm 1), the desired return-to-go $\tilde g_1$ can be assigned either as the highest return in dataset, or a pre-specified return to achieve according to the task. Our definition for the training and evaluation processes of RCSL aligns with previous RCSL literature [1-3].
>
> [1] David Brandfonbrener, Alberto Bietti, Jacob Buckman, Romain Laroche, and Joan Bruna. When does return-conditioned supervised learning work for offline reinforcement learning? Advances in Neural Information Processing Systems, 35:1542–1553, 2022.
>
> [2] Scott Emmons, Benjamin Eysenbach, Ilya Kostrikov, and Sergey Levine. Rvs: What is essential for offline rl via supervised learning?, 2022.
>
> [3] Lili Chen, Kevin Lu, Aravind Rajeswaran, Kimin Lee, Aditya Grover, Misha Laskin, Pieter Abbeel, Aravind Srinivas, and Igor Mordatch. Decision transformer: Reinforcement learning via sequence modeling. Advances in neural information processing systems, 34:15084–15097, 2021.

---

> ### Author Response · Authors · 2023-11-21
> **Follow-Up**
>
> Dear Reviewer,
>
> Thank you for your time and efforts in reviewing our work. We have provided detailed clarification to address the issues raised in your comments. If our response has addressed your concerns, we would be grateful if you could re-evaluate our work.
>
> If you have any additional questions or comments, we would be happy to have further discussions.
>
> Thanks,
>
> The authors

---

> ### Comment · Reviewer_YXKR · 2023-11-23
> **Thank you for your detailed response.**
>
> If my understanding is correct, what this paper does is to propose a preprocessing steps for RCSL, which is essentially a data augumentation prodedure based on a learnt model. In this way, the problem that "RCSL CANNOT PERFORM TRAJECTORY-STITCHING" could potentially be addressed. But I'm still curious about why other offline methods, e.g., Decision Transformer cannot benefit from those roll outs.  In other words, I think that the close connection between the proposed method and RCSL is not well revealed in this paper although it shows that such combination indeed yields good empirical results. Besides, employing rollouts from a model is a popular way for model-based offline RL, and in this sense, the novelty of the proposed method could be further clarified.

---

> > ### Author Response · Authors · 2023-11-23
> > **Response to Reviewer YXKR**
> >
> > Thank you for response. We address your questions below.
> >
> > > **Why other offline methods, e.g., Decision Transformer cannot benefit from those rollouts?**
> >
> > Decision Transformer (DT) belongs to an instance of RCSL framework [1,2], and thus model-based rollouts + DT (referred to as MBDT) is included in our MBRCSL framework.
> >
> > In fact, DT also benefits from rollouts. We compared DT, MBDT and MBRCSL (The RCSL output policy is implemented with MLP network, as described in Section 5.1.) on Point Maze, as shown in the table below:
> >
> >
> > | Task | DT | MBDT | MBRCSL (ours) |
> > | -------- | -------- | -------- | -------- |
> > | Pointmaze    |  57.2    | 79.8     | **91.5** |
> >
> > We can see that MBDT significantly DT. However, MBRCSL still achieves higher averaged return than MBDT. This result is due to architectural difference in RCSL output policy and aligns with [2], in which the authors claim that RCSL policy with MLP network can match performance of DT in some environments. We will also investigate the RCSL output policy architecture in MBRCSL framework as a future direction.
> >
> >
> > [1] David Brandfonbrener, Alberto Bietti, Jacob Buckman, Romain Laroche, and Joan Bruna. When does return-conditioned supervised learning work for offline reinforcement learning? Advances in Neural Information Processing Systems, 35:1542–1553, 2022.
> >
> > [2] Scott Emmons, Benjamin Eysenbach, Ilya Kostrikov, and Sergey Levine. Rvs: What is essential for offline rl via supervised learning?, 2022.
> >
> > > **Employing rollouts from a model is a popular way for model-based offline RL, and in this sense, the novelty of the proposed method could be further clarified.**
> >
> > As far as we know, our MBRCSL framework is the *first* model-based offline RL method that can avoid Bellman completeness requirements. Previous model-based offline RL methods such as COMBO, MOPO and MOReL require estimation of the value function or $Q$-function, which involves dynamic programming, so that they often diverge in practice due to the absence of Bellman completeness in the function classes considered.
> >
> > In contrast, MBRCSL utilizes *no* dynamic programming. Instead, it simply uses supervised learning in all processes (dynamics model learning, behavior policy learning and RCSL output policy learning). As a result, it avoids the Bellman completeness requirement and converges under more relaxed conditions.

---

> > > ### Comment · Reviewer_YXKR · 2023-11-23
> > > **Thanks for your clarification**
> > >
> > > One more question, do you compare the proposed method with Elastic Decision Transformer (EDT), an alternative method that also aims to address the trajectory-stitching issue?

---

> > > > ### Author Response · Authors · 2023-11-23
> > > > **Thanks for the pointer**
> > > >
> > > > The motivation of EDT [1] is to automatically select the optimal context length for DT by estimating the maximum returns under given context lengths. EDT also avoids Bellman completeness requirements as it only uses supervised learning.
> > > >
> > > > Compared with our MBRCSL method, the main limitation of EDT is that it cannot do trajectory stitching in *all* cases. This is because the maximal value estimator $\tilde R$, according to the motivation of EDT, is learned as the maximum among a collection of return-to-gos in the dataset (cf. Eq. (1) of [1]). In other words, it cannot generate the **optimal** return if it is not present in the trajectory.
> > > >
> > > > We give a counter-example similar to Figure 3 of [1]. The MDP has 5 states $\{A,B,C,D,E\}$ with the initial state being $A$. The dataset has 2 trajectories: 1) $A \to D \to B \to E$; 2) $A\to B \to C$. The reward is : $r(A\to D) = 0.5$, $r(D\to B)=0$, $r(B\to E) = 0$, $r(A\to B) = 0$, $r(B\to C) = 1$. We omit the action space for simplicity. The optimal trajectory should be $A\to D \to B \to C$ with return 1.5, while the maximum return in dataset is 1.
> > > >
> > > > At the first step, EDT will transit to state $B$, i.e., following the second trajectory in dataset. This is because the training of maximal value estimator $\tilde R$ only has access to return-to-gos computed by dataset trajectories, and the second trajectory has a higher return. Consequently, EDT can only reach return 1 instead of 1.5.
> > > >
> > > > In contrast, MBRCSL is able to do trajectory stitching in all cases. For instance, in the example above, MBRCSL will generate rollout trajectories 1) $A \to D \to B \to E$; 2) $A \to D \to B \to C$; 3) $A\to B \to C$; 4) $A\to B \to E$, in which the optimal trajectory is included and will be extracted by the RCSL output policy.
> > > >
> > > > Admittedly, EDT does address part of the trajectory stitching problems in RCSL, like the example in Figure 3. of [1]. In that case, EDT finds the optimal trajectory  because the first action in the dataset already corresponds to expert policy (which is the only choice in that example).
> > > >
> > > > The official implementation of EDT was released at Oct 12th 2023, which was after the ICLR submission deadline. We are happy to add a comparison with EDT in the final version.
> > > >
> > > > [1] Wu, Yueh-Hua, Xiaolong Wang, and Masashi Hamaya. "Elastic decision transformer." Preprint arXiv:2307.02484 (2023).

---

### Author Response · Authors · 2023-11-20
**Common Response: Changes to paper**

**We thank the reviewers for their careful reading and constructive feedback. To address the questions raised in the reviews, we made the following changes to our paper:**

1. More environments: We added results of "ClosedDrawer" and "BlockedDrawer", two simulated robotic tasks besides "PickPlace", in Table 2 of Section 5.
2. More baseline: We added results of MOReL [1], another model-based offline RL algorithm, for Point Maze task in Table 1 of Section 5.

   [1] Kidambi, R., Rajeswaran, A., Netrapalli, P., & Joachims, T. (2020). Morel: Model-based offline reinforcement learning. Advances in neural information processing systems, 33, 21810-21823.
3. Ablation studies: We added a complete ablation study in Appendix C about the effects of dynamics model, behavior policy, rollout dataset size, data distribution and output policy on MBRCSL performance.
4. Empirical studies of Bellman completeness: We added discussions in Appendix D to show that performance of CQL does not increase with model size on Point Maze task, which implies that Bellman completeness cannot be satisfied by simply increasing model size.
5. Experiments on comparison between RCSL versus $Q$-learning: In Appendix A.3, We additionally tested the performance of RCSL and CQL on Point Maze given expert dataset under different model sizes, to validate the claim of Section 3.1 that RCSL outperforms DP-based methods in deterministic environments.

We provide additional clarifications, explanations and discussion in the per-reviewer responses.

---

### Meta-Review · Area_Chair_v3Ux · 2023-12-08

**Metareview:**

The paper studies a family of methods for reinforcement learning called return-conditioned supervised learning (exemplified by Decision Transformers and other recent works). The paper theoretically characterizes conditions when rcsl can work without requiring a Bellman-completeness assumption, and instantiate practical scenarios where stitching together trajectory segments from offline data can cover the optimal policy trajectories. Finally the paper derives a method (MBRCSL) and compares it to state-of-the-art offline RL algorithms, finding it to be competitive and outperforming other rcsl techniques like Decision Transformers. The reviewers raised some concerns about experimental evaluation and theoretical characterization that the authors addressed comprehensively during the discussion phase.

**Justification For Why Not Higher Score:**

Many additional experiments were added during the discussion phase and all of them could not be scrutinized carefully (e.g. in one experiment, Decision Transformer did not get any non-zero rewards -- was it a bug in the implementation or a known failing of the method?). There was also some disagreement among the reviewers about the significance of the theoretical contribution given what was already known from the Brandfonbrener et al paper on RCSL recently.

**Justification For Why Not Lower Score:**

All the reviewers agree that the paper's proposed solution for offline RL is promising, and the theory sheds light on at least some situations when RCSL can work well.

---

### Decision · Program_Chairs · 2024-01-16

Accept (poster)